# DSDF: Coordinated look-ahead strategy in multi-agent reinforcement learning with noisy agents

## Abstract

Existing methods of Multi-Agent Reinforcement learning, involving Centralized Training and Decentralized execution, attempt to train the agents towards learning a pattern of coordinated actions to arrive at optimal joint policy. However, during the execution phase, if some of the agents degrade and develop noisy actions to varying degrees, the above methods provide poor coordination. In this paper, we show how such random noise in agents, which could be a result of the degradation or aging of robots, can add to the uncertainty in coordination and thereby contribute to unsatisfactory global rewards. In such a scenario, the agents which are in accordance with the policy have to understand the behavior and limitations of the noisy agents while the noisy agents have to plan in cognizance of their limitations. In our proposed method, Deep Stochastic Discount Factor (DSDF), based on the degree of degradation the algorithm tunes the discount factor for each agent uniquely, thereby altering the global planning of the agents. Moreover, given the degree of degradation in some agents is expected to change over time, our method provides a framework under which such changes can be incrementally addressed without extensive retraining. Results on benchmark environments show the efficacy of the DSDF approach when compared with existing approaches.

## 1 Introduction

Multi-agent reinforcement learning (MARL) has been applied to wide variety of works which involve collaborative behavior such as traffic management (Chu et al., 2019), power distribution (Nasir & Guo, 2019), fleet management (Lin et al., 2018) *etc*. There are different methods to encourage this collaboration among agents. While a set of algorithms focus on learning centralized policies (Jaques et al., 2019; Moradi, 2016), some learn decentralized policies (Zhang et al., 2018). To improve the performance of the decentralized policies, some works leveraged centralized training while learning these policies (Tan, 1993; Rashid et al., 2018; Mahajan et al., 2019; Peng et al., 2021). In literature these methods are known as *centralized training and decentralized execution* (CTDE) methods.

Subsequently, there are many CTDE methods proposed for obtaining collaborative multi-agent policy. These include preliminary methods like IQL (Tan, 1993; Xu et al., 2021) which has challenges dealing with non-stationarity and then extends to more recent methods like COMA (Foerster et al., 2018), VDN (Sunehag et al., 2017), QMiX (Rashid et al., 2018), MAVEN (Mahajan et al., 2019), QTRaN (Son et al., 2019), FACMAC (Peng et al., 2021) *etc*. Some variants of these methods can be found in (Xu et al., 2021; Rashid et al., 2020). All these approaches assume the agents behave exactly the way the policy instructed it. However in some cases, the agents can behave inconsistently *i.e.*, sometimes they can execute actions different from the actions given by the policy (Dulac-Arnold et al., 2019). The degree of inconsistency can be different for different agents *i.e.*, the probability of executing an action different from the one given by the policy, can vary. In the rest of the paper, we refer to such agents as *noisy/degraded* agents. This is an expected phenomenon in robotic manufacturing for Industry 4.0 where agents (machines *etc*.) may undergo wear and tear and subsequently are degraded which can result in noisy actions. More details on the Industry 4.0 use case and relevance is given in appendix D. To explain it better let us consider the following intuitive example.

Consider the case of a soccer match wherein one or more players are injured and hence their movement and precision are impacted. Let us assume that these players cannot be replaced and all the players have similar skills. So the intuitive team strategy would be to let the injured players operate within a small radius and just perform small distance passes while leaving the dribbling/running to other fit players. Effectively, injured players take simpler short-term objectives

(passing the ball) while other fit players take relatively longer and more complex goals (like dodging and running with the ball for longer distances). This, in turn, means all the players would need to refactor their look-ahead strategy and re-tune the respective policies to maximize the overall joint policy reward. Such refactoring would also be essential in a robotic environment, like robots coordinating in space where they cannot be replaced readily or even near home in an Industry 4.0 use case for assembly line manufacturing, where some of the robots may get degraded due to wear-n-tear over time. Given maintenance or replacement for robots is a costly process (especially so for space explorations), the concept described here could enable the continuation of the manufacturing process as one robot adjusts to other's shortcomings. This, in turn, saves cost, increases efficiency, and partially contributes to reducing the carbon footprint by increasing the lifetime of a robot.

In the context of RL, intuitively, the refactoring of strategy can be achieved by adjusting the discount factor for each robot. For example, noisy (degraded) agents can use lower discount factor *i.e.*, they can plan for short-term rewards (myopic) whereas the good and accordant agents should use higher discount factor *i.e.*, they can plan for long-term rewards. However, the choice of tuning of the discount factor for each agent is non-trivial in a multi-agent scenario. Since the discount factor of one agent will have an effect on another agent's strategy, such tuning can only be done based on a unified representation of the state of the environment and the differential capabilities of other agents.

Specifically, let us consider an agent with *Degradation Value* of 0.1, *i.e.*, with a probability of 0.9, it will correctly follow the action given by policy and perform any other noisy action with a probability of 0.1. Here $P$(Correct action) = 0.9, $P$(Noisy action) = 0.1. Assuming an episode has 10 steps, the probability of executing at-least 80% of steps correctly per policy, in the episode, is given by:

$P$(Correct action)$^8$ * $P$(Noisy action)$^2$ + $P$(Correct action)$^9$ * $P$(Noisy action) + $P$(Correct action)$^{10}$ = 0.39.

If the episode length increases to 20, the corresponding probability drops to 0.14 while for an episode length of 5, the probability increases to 0.65. Figure 1 shows the diminishing probability of "at least 80% correct steps per episode" with increase in length of episodes. Given the success of the collaboration between agents in a CTDE framework depends upon successfully executing learned joint actions, any deviation from planned joint actions may result in upsetting the collaboration and thereby limiting the global reward. Such a deviation is caused by one or a few such noisy agents, but it affects the overall ability to execute the joint coordination. More the probability of deviation from policy, more the risk of disruption to joint coordination. Hence reducing the probability of such deviation by shortening the look-ahead horizon for noisy agents may be a reasonable trade-off. Such an arrangement makes the noisy agents near-sighted and thereby encourages them to create short-term plans

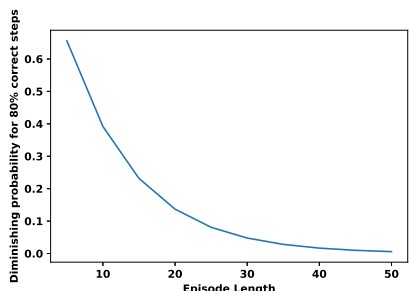

Figure 1: Final degradation probability obtained at the end of episode vs episode length.

which has a lower probability of deviation than a longer-term plan. Correspondingly, the accordant agents need to understand that the few agents would have a short-term horizon and can re-plan their strategy accordingly, sometimes compensating by extending their horizon.

The proposed framework can be also seen as robust MARL where agents are designed to handle noisy environments (Luo et al., 2020; Zhang et al., 2020). In these works, the noise is in the form of corrupted observations using which agent decides on the best actions. However, the proposed work assumes that the actions of the agents are corrupted (discrete in nature). To the best of our knowledge, we have not come across any such work in the literature that deals with improving global coordination when the actions are corrupted.

The problem described here has also been viewed from context of Ad-Hoc Teaming(AHT) (Mirsky et al. (2022)) which is a popular framework for centralized coordination that employs pre-trained self-interested agents to solve a coordinated task, where a "Learner" agent learns to set goals or incentives to drive the coordination among worker agents. AHT excels in circumstances where unseen agents may arrive in the system or may even change their existing type. The change in the existing type could be viewed as a change in the degradation factor in our problem context (Ravula et al., 2019). However, in our work, we assume a decentralized execution framework whereby there is no central entity to communicate a changed plan of action given certain agents has degraded. In the AHT scenario, we will need a centralized "Learner" with preferably global visibility which the given environments may not realistically

offer. Also typesetting all the variations of degraded actions may be a challenge. Finally, AHT Learners may not be able to influence the agents to change basic attributes of their policy i.e to be far or near-sighted as the agents have their own pre-existing policies.

Our proposed method may also be compared with concepts of generalization in reinforcement learning (Hansen & Wang, 2021; Cobbe et al., 2019) However in our context the environment or the accompanying dynamics does not change. Rather the action changes from the prescribed policy due to the induced noise. Hence this method, with some changes, could also be used for adapting to unseen agents with different types of action selection policies. At this point we do not pursue this aspect and have left it for future work.

In the recent past, there have been some workarounds tuning the discount factor in the context of single-agent reinforcement learning (Xu et al., 2018). However, to the best of our knowledge, there exists nothing in the context of MARL. Hence in this work, we propose an efficient representation that could help all agents to understand the effective discount factor for each of them. We call our approach the *Deep Stochastic Discounted Factor* (DSDF) method, which predicts the discount factor based on a hypernetwork. The method leverages the observation space of the agents and the state of the environment. The proposed method computes an unique discount factor for each agent simultaneously and promotes effective cooperation in the context of the degraded scenario. Once the right discount factors are predicted and a joint look-ahead strategy is devised, the next challenge comes in the form of continuous degradation. The degradation of agents due to wear-n-tear is a continuous process and a joint policy trained once may not be valid afterwards. Hence it may seem essential to retrain both learning representation and the joint policy. However, our results illustrate that the learning representation in form of the hypernetwork, which provides the right combination of discount factors, will remain valid without the need for retraining. The discount factor values for each agent would however change based on the extent of degradation. Only incremental training for the agents' network would be needed. Therefore our work also suggests the thresholds at which such incremental training is necessitated and provides an incremental method for the same.

The proposed approach of handling noisy agents is tested on four benchmark environments, where the action space is discrete in three (i) SMAC (Samvelyan et al., 2019b), (ii) Google Football (Kurach et al., 2020), (iii) lbForaging (Papoudakis et al., 2020) and continuous in (iv) Water World (Gupta et al., 2017) . Different scenarios are tested in each of the environments. Results on these environments shows the efficacy of the proposed approach in terms of higher average reward when compared with existing methods.

## 2    Background

In this work, we assume a fully cooperative multi-agent task which is described with a decentralized partially observable Markov decision process (Dec-POMDP) which is defined with a tuple $G = \langle S, A, P, r, Z, O, N, \gamma \rangle$ where $s \in S$ describes the state of the environment (Bernstein et al., 2002). At each step in time, each agent $i$ out of the $N$ agents will take an action $a_i \in A$ and for all the agents the joint action is represented by $\mathbf{a} = [a_1, a_2, \cdots, a_N]$. Due to the joint action applied, the system will transit to state $s'$ with probability $P(s'|s, \mathbf{a}) : S \times \mathbf{A} \times S \longrightarrow [0, 1]$. All the agents share a common reward function $r(s, \mathbf{a} : S \times \mathbf{A} \in \mathbb{R})$ and each agent has a different discount factor values $\gamma^i \in (0, 1)$ using which the future states are weighted.

We consider the environment is partially observable in which the agent draws individual observation $z \in Z$ according to an observation space $O(s, a) : S \times i \longrightarrow \mathbf{Z}$, for agent $i$. Each agent can have their own observation history $\tau^i \in \tau : (Z \times A)$ which influences the underlying stochastic policy $\pi^i(a^i|\tau^i)$. The joint policy $\pi$ has a joint action-value function $Q^\pi(s_t, \mathbf{a}_t) = E_{s_{t+1:\infty}, \mathbf{a}_{t:\infty}}[R_t|s_t, \mathbf{a}_t]$, where $R_t = \sum_{i=0}^{\infty} f(.)^i r_{t+i}$ and $f(.)$ is the function to predict discount factors.

## 3    Proposed method

At the start of the execution, all agents are conformal and accordant with policy. After a while, one/more agents start degrading. At some point, performance degradation is observed in the form of diminished global reward. Once the degradation in performance exceeds an user-defined threshold (which is dependent on the sensitivity of the application), the joint policy needs to be retrained. The non-trivial determination of the optimal individual $\gamma^i$ is obtained

using a learning representation, which is explained in greater detail in subsequent paragraphs. Two approaches have been followed here, one is retraining the agent policy from scratch, and another is retraining the policy incrementally using transfer learning. Subsequently, agents may continue to degrade and the above method is followed iteratively. It is important to note that, the DSDF Learning Representation, once trained, need not be retrained in either of the approaches. Only the policy network needs to be retrained for such degraded scenario. The method thus circumvents the need for continuous learning during execution, which is outside the scope of the paper.

The proposed algorithm and the training approach is explained in the context of QMIX (Rashid et al., 2018) since QMIX is one of state of art collaborative mechanisms. However, DSDF, in principle, can be extended to any other collaborative mechanism which leverages CTDE.

In QMIX we compute the value function (also known as utility function) for agent $i$ out of all $N$ agents as $Q_i(\tau^i, a^i)$, where $\tau^i$ is the history of observation space and $a^i$ is the action for agent $i$. The agent utility values $Q_i(\tau^i, a^i)$ are obtained by sending the observation spaces $(\mathbf{o}^i, a^i)$ through a network with parameters $\theta_i$, where $i$ is the agent index out of $N$ agents. The obtained utility values $Q_i(\mathbf{o}^i, a^i) \ \forall \ i = 1, \cdots, N$ are mixed using a mixing network to obtain combined mixing value $Q_{\text{tot}}(\tau, \mathbf{a})$. Now agent $i$ network, $\theta_i$ is updated by total value function $Q_{\text{tot}}$ instead of local utility function $Q_i$. The advantage here is that each agent network will have information on the reward obtained and also indirectly about the other agent's performance. In this way, we can incorporate other agents' information without actually collecting them and are able to arrive at a joint policy.

As discussed, the discount factor for each agent is computed based on the current observation and also the global state of the environment. We propose a method known as DSDF which is a hypernetwork-based learning representation to compute the appropriate discount factors for each agent. For this, a fully connected neural network is used which will output the discount factor values $\gamma^i$ for each agent $i$. In both the above cases, the computed discount factor is fed to the individual utility networks. Now, both the utility networks and mixing network (hypernetwork to be exact) are updated using the different discount factors (predicted using the discount factor network explained earlier) for each agent. This forms the crux of the proposed DSDF and also the iterative penalization method.

In this work, the noisy agents are assumed to be stochastic with a factor $\beta$ i.e.

$$a^i = \begin{cases} a^i & \text{with probability } 1 - \beta \\ \text{randint}(1, |A^i|) & \text{with probability } \beta \end{cases} \tag{1}$$

where $a^i$ is the action suggested by the policy for the agent $i$ and $|A^i|$ is set of actions for the agent $i$. From the expression, it is understood that any action from the entire action space has a uniform probability to replace the action suggested by the policy. However, in many situations it may not be correct i.e. every action will have a different probability when replacing the action suggested by the policy. For example, assume there is a Mars rover that can take actions to go in any of the 8 directions. Also, it is assumed the action suggested by the policy is to 'left action'. Now, due to the stochasticity of the agents in our approach, taking any action which contains left like the top left, and bottom left is of high probability than other actions. Hence in this work, we propose to use the following formulation of noisy agents which is more realistic

$$a^i = \begin{cases} a^i & \text{with probability } 1 - \beta \\ \text{randint}(1, |A^{is}|) & \text{with probability } \beta \end{cases} \tag{2}$$

where $A^{is}$ is the subset of actions defined for the action suggested by the policy $a^i$.

In the above two noise formulations, it is assumed that the action space is discrete-one. However, many real-time systems have continuous action spaces and hence we need to formulate the noise model for the continuous action spaces. For this case, the noise model is formulated as

$$a^i = a^i \pm (a^i * \beta) \pm \epsilon^i \tag{3}$$

where $a^i$ is the action given by policy, $\beta$ is the degradation factor for the agent and $\epsilon^i \sim \mathcal{N}(0, \sigma_{\epsilon^i}^2)$ added to the agent to remove the repetition of the same value. $\sigma_{\epsilon^i}^2$ is the variance of the noise $\epsilon^i$.

Notably, the value of $\beta$ differ from agent to agent and could change during the lifetime of the agent.

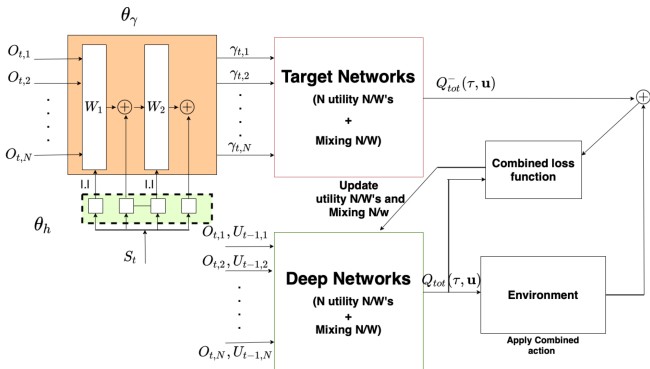

Figure 2: Proposed DSDF approach. The individual target $Q$-values are multiplied with predicted discount values and added to the reward obtained from the environment. Finally, this value is used to update the $N$ utility networks and mixing network (In the case of QMIX approach)

The noise formulation in (1), (2) and (3) is different and in addition from that of general policy exploration. The proposed approach assumes these two noise components are present (i) general exploration by the policy during training and (ii) noisy actions which persist during execution. Due to the first component, the policy trained will converge to exploitation obeying GLIE property (Singh et al., 2000).

### 3.1 Proposed methods to calculate appropriate $\gamma$ for all the agents

In this work, we propose a DSDF approach to compute the appropriate $\gamma$ for all the agents. For comparison, we also came up with a method called iterative penalization to arrive at the discount factor. In this iterative penalization method, it is assumed that the agents have the knowledge of the applied action in the environment. While this may not be a realistic assumption, but this serves for purposes of bench-marking.

#### 3.1.1 Iterative penalization Method

In this method, the discount factor is set to 1 for all the agents at the start of the episode. If the action executed by the agent $i$ is different from that of the action given by the policy, we will penalize the discount factor for the agent $i$ by a value $P$. At every time step, with every mismatch in the action taken from policy, the discount factor is decreased by a value $P$. We do the approach for multiple episodes to arrive at an average discount factor for each agent. Now, this averaged discount factor (for each agent) is used to compute the utility function. Finally, the regular QMIX approach is applied to update the utility networks and mixing networks.

The proposed iterative penalization method to estimate the discount factor with the underlying QMIX method is explained in algorithm 1. The penalization value $P$ should decrease with time steps just like we do in the exploration factor in the $\epsilon$-greedy method. The choice of optimal value for $P$ can be itself posed as an optimization problem which is out of the scope of this paper. However, the method proposed in Section 3.1.2 does not require any hyperparameter selection.

#### 3.1.2 Deep Stochastic Discounted Factor (DSDF) Method

In this case, we propose a method to compute the appropriate $\gamma^i, \quad i = 1, \cdots, N$ using a trained network. The proposed method is shown the Figure 2.

The idea is to utilize the agent local observations $o_t^i$ and global state $s_t$ at time instant $t$ of the environment to compute the discount factor for individual agents. The reason is explained below:

Since it is assumed the underlying collaborative mechanism works for accordant agents, the only issue behind poor global return is the degradation of some agents and the noise in action selection. Each agent has their perspective of the global state in form of local observations. Hence local observations will have an idea of the noise in the degraded

---

Algorithm 1: Proposed iterative penalization method to estimate discount factor

---

**Require:** No. of episodes to compute discount factor $E_1$, No. of episodes to train the agents $E_2$, $\gamma = \mathbf{0}^N \times 1$

    **for** e = 1:$E_1$ **do**
        Initialize $\gamma^i = 1 \ \forall \ i = 1, \cdots, N$
        **for** i = 1:N **do**
            **if** mismatch in action executed and action defined by policy **then**
                Update $\gamma^i = \gamma^i$ - $P$
            **end if**
        **end for**
        Update $\gamma^i = \gamma^i/E$
    **end for**
    **for** e = 1:$E_2$ **do**
        Update the utility networks and mixing network for every $B$ samples with $\gamma_i$ value for each agent.
    **end for**

---

agents and we need to estimate the discount factor depending on the local observations and also the global state of the environment.

Returning to the main discussion, the local observations of agents are used to estimate the discount factors using a fully connected network $\theta_\gamma$. However, additional information about the global state is required. Since the local observations and global state are in different scale, we cannot combine together in the same network. Also, we have a non-negativity constraint on the $\gamma$ values which is enforced by training the $\theta_\gamma$ using the concept of hypernetwork as described in (Ha et al., 2016). The local observations of all $N$ agents are sent to the network $\theta_\gamma$ which will compute the discount factor values $\gamma^i, \ i = 1, \cdots, N$. We utilize the hypernetwork $\theta_h$ to estimate the network weights $\theta_\gamma$. The training process to update the $\theta_h$ is explained below.

The $\theta_h$, $\theta_u$ (utility networks), and $\theta_m$ (mixing hypernetwork) are interlinked with each other. The general loss function of QMIX is

$$\mathcal{L}(\theta) = \sum_{t=1}^{B} \left(y_t - Q_{tot}(\tau, \mathbf{a}, s : \theta)\right) \tag{4}$$

where $\theta$ is the set of parameters of $N$ utility agents $(\theta_t^i, i = 1, \cdots, N)$ and mixing hypernetwork $\theta_m$, computed over $B$ episodes. Now, if we expand $y_t$

$$y_t = r + \gamma \ \max_{a'} \ Q_{tot}(\tau', \mathbf{a}', s' : \theta^-) \tag{5}$$

Now, instead of using a single $\gamma$ value, we will take the $\gamma$ inside the equation and use the mixing network function to calculate the value of $Q_{tot}$

$$y_t = r + \max_{a'} \ g(\mathbf{a}', s', \gamma_i Q_{a,i}, \theta_{tot}) \tag{6}$$

Here $g(.)$ is the mixing network architecture which is parametrized by $\theta_{tot}$, $Q^i$ is the individual utility function of agent $i$. Now, we will replace the $\gamma^i$ with the output of the network $\theta_\gamma$. The replaced equation is

$$y_t = r + \max_{a'} \ g(\mathbf{a}', s', f_\gamma(o_t^1, \cdots, o_t^N, \theta_\gamma)Q_{a,i}, \theta_{tot})$$
$$= r + \max_{a'} \ g(\mathbf{a}', s', f_\gamma(o_t^1, \cdots, o_t^N, f_h(\theta_h, s'))Q_{a,i}, \theta_{tot}) \tag{7}$$

where $f_\gamma$ is the discount factor hyper network which is parametrized by $\theta_\gamma$ and $f_h$ is the hypernetwork function which is parametrized by $\theta_h$.

Replacing the value of $y_i^{tot}$ from (7) with that of (4) we obtain the loss function as

$$\mathcal{L}(\theta, \theta_h) = \sum_{t=1}^{B} \left(r_t + \max_{a'} g(\mathbf{a}', s', f_\gamma(o_t^1, \cdots, o_t^N, f_h(\theta_h, s')) \cdot Q^i, \theta)\right)$$
$$- Q_{tot}(\tau, \mathbf{a}, s : \theta) \tag{8}$$

---

**Algorithm 2:** DSDF method to estimate discount factor with QMIX (training from scratch)

---

**Require:** Initialize parameter vector $\theta_h$, hypernetwork parameters and $\theta$ (agents utility networks, maxing network, hyper network), Learning rate $\leftarrow \alpha_\gamma$ and $\alpha_\theta$, $\mathcal{B} \leftarrow \{\}$

**Require:** step = 0, $\theta^- = \theta$

1: **while** step < step$_{max}$ **do**
2:      $t = 0$, $s_0$ = Initial state
3:      **while** $t \neq$ terminal and $t <$ episode limit **do**
4:          **for** each agent $i$ **do**
5:              $\tau_t^i = \tau_{t-1}^i \cup \{(o_t, a_{t-1})\}$
6:              $\epsilon$ = epsilon-schedule (step)
7:              $a_t^i = \begin{cases} \underset{a_t^i}{\text{argmax}} \ Q(\tau_t^i, a_t^i) & \text{with probability } 1 - \epsilon \\ \text{randint}(1, |A^i|) & \text{with probability } \epsilon \end{cases}$     ▷ Policy exploration present only during training
8:              Introduce noisy actions using the noise model either in (2) or (3).     ▷ Noisy actions present during training and execution
9:          **end for**
10:          $s_{t+1} = p(s_{t+1}|s_t, \mathbf{a}_t)$
11:          $\mathcal{B} = \mathcal{B} \cup \{(s_t, \mathbf{a}_t, r_t, s_{t+1}\}$
12:          $t = t + 1, step = step + 1$
13:      **end while**
14:      **if** $|\mathcal{B}| >$ batch-size **then**
15:          b $\leftarrow$ random batch of episodes from $\mathcal{B}$
16:          **if** $||\theta_{h_{k+1}} - \theta_{h_k}||_2^2 \geq \epsilon$ **then**
17:              Update $\theta_h = \theta_h - \alpha_\gamma \nabla_{\theta_h}(\Delta Q_{tot})^2$
18:              Update $\theta_\gamma = f_\gamma(O, \theta_h)$, where $O$ is the set of observations for all agents in the sampled batch.     ▷ Updating the discount factor network
19:          **end if**
20:          Update $Q_{tot}$ using the latest updated $\theta_\gamma$.
21:          Update $\theta = \theta - \alpha_\theta \nabla_\theta(\Delta Q_{tot})^2$
22:      **end if**
23:      **if** update-interval steps have passed **then**
24:          $\theta^- = \theta$
25:      **end if**
26: **end while**

---

There are two unknown parameters in the above equation (i) $\theta$, (ii) $\theta_h$. Since the parameters are interdependent on each other i.e. $\theta_h$ on $\theta$ and vice-versa, we need to solve them iteratively. For every $B$ samples, first, we will update the hyper network $\theta_h$ for fixed $\theta$ and then update $\theta$ for the computed $\theta_h$. So at each step, we update both $\theta_h$ and $\theta$ iteratively.

An important point to note here is that $\theta$ needs to be updated for every batch of samples as each batch will have new information on the environment. However, in the real world, the degree of degradation might become stationary after some time (an injured soccer player or a robot degrades till a point but holds on to the level of degradation). Consequently, the noise in the degraded agents may remain constant for a while. Hence the $\theta_\gamma$ network will converge after some iterations once it has figured out the right values of the discount factor. Hence at that point, there is no need to update the $\theta_\gamma$ and $\theta_h$. However, it should be noted that the discount factor value will change depending on the local observations and the state of the environment during the execution. Please refer to Figures 3a, 3b and 3c, where the training of $\theta_\gamma$ is done on one experiment (one scenario of degradation) and the other experiments used the same $\theta_\gamma$.

We can also apply the proposed DSDF approach to the case where all the agents are accordant. In this case, we expect it will give improved performance when compared with existing methods (like QMIX) as it will dynamically tune the discount factor $\gamma$ for each agent, based on its local observation also, rather than using constant $\gamma$ for all the agents.

---

Algorithm 3: DSDF method to estimate discount factor with QMIX (updation of agents for continuous degradation)

---

**Require:** exec_step = E, Threshold = T
 1: Use Algorithm 2 to obtain $\theta_\gamma$ and $\theta$.
 2: **for** step in 1:E **do**
 3:     Execute the trained agents on the environment and monitor the performance.
 4:     **if** Performance degradation for two consecutive episodes $<$ T **then**
 5:         Update the $\theta$ using the algorithm 2 with $\theta_h$ converged.
 6:     **end if**
 7: **end for**

---

The proposed DSDF method to estimate the learning representation along with the agents' network updation/training is given in Algorithm 3.

The choice of threshold to decide on retraining/updating is important but it is application dependent. Ideally, different experiments have to be performed to validate the effect of degradation, thereby the appropriate threshold value can be determined. However given this is application-centric ( eg. precision manufacturing is more sensitive and has lower thresholds than warehouse robots), it is left outside the scope of this paper.

# 4 Results and Discussion

The proposed approach is validated on four different benchmark environments (i) SMAC (Samvelyan et al., 2019b), (ii) Football (Kurach et al., 2020), (iii) Modified lbForaging (Papoudakis et al., 2020) and (iv) Water World (Gupta et al., 2017). To induce the degradation in agents, a noisy component $\beta$ is added which will decide whether to execute the action given by the policy or to take a random action from a set of valid actions. Note that this is different and in addition to the $\epsilon$-greedy exploration-exploitation strategy used in the training phase.

The agents in SMAC and Football environments need to exhibit collaboration among themselves while effectively competing against the enemy. The lbForaging presents a cooperative scenario but it is possible to observe individual reward attributions and thereby the behavioral change in each agent. In all these three environments, the action space is discrete whereas the Water World presents an environment where the action space is continuous.

## 4.1 Description of the Environments

### 4.1.1 SMAC Environment

SMAC (Samvelyan et al., 2019b) simulates the battle scenarios of a popular real-time strategy game StarCraft II. Similar state space, observation space, action space, and reward are used as with the (Rashid et al., 2018). The results were compared on three different sets of maps (i) 2s_3z, (ii) 8m and (iii) 1c_3s_5z.

The proposed approach was tested in two different scenarios in this environment (i) Experiments with different noise values for agents and training the joint policy for each experiment from scratch and (ii) Continuous degradation of agents whereby beyond a threshold of performance degradation, the joint policy is updated from the previous state. In both these scenarios, the discount factor learning representation was trained only once. In the case of scenario 1, the learning representation was trained for the first experiment and the network is reused for all remaining experiments. For scenario 2, the learning representation was trained for the case where all the agents are accordant and it is reused for all subsequent cases where agents become noisy. Scenario 1 demonstrates the efficacy of our method on non-correlated scenarios whereas scenario 2 demonstrates a feasible way for implementation in real-world environments where degradation may be continuous and monotonic.

**Scenario 1**: **For each map, 10 different sets of experiments were performed with the different number of noisy agents in each experiment and with varying degrees of degradation values.** For each experiment, the probabilistic degradation values $\beta$ were assigned to the agents, as shown in Figures 3a, 3b, and 3c.

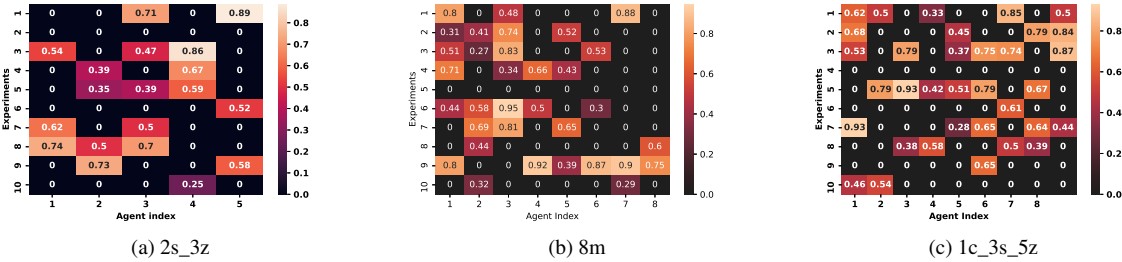

(a) 2s_3z       (b) 8m       (c) 1c_3s_5z

Figure 3: Degradation values of agents considered in each experiment for SMAC

Table 1: Subset of actions to be replaced instead of action suggested by policy

| Action Suggested by policy | Subset of actions need to be replaced |
|---|---|
| Top | [Top, Top left, Top right] |
| Bottom | [Bottom, Bottom left, Bottom right] |
| Left | [Left, Top left, Right, Bottom left] |
| Right | [Right, Top Right, Bottom Right, Left] |
| Long pass | [Long pass, High pass, Short pass] |

Here $\beta$ means the agent will perform actions given by the policy with $1 - \beta$ probability and perform random actions with $\beta$ probability. The degradation value of 0 implies the agent is good or accordant with policy. The experiments also included the case where all the agents are accordant to demonstrate the applicability across scenarios.

**Scenario 2**: In this approach, we consider a sequential and continuous degradation and update Q-networks of agents only when global performance degrades beyond a threshold. **Specifically, we update the agents' networks from the previous state instead of training from scratch.**

### 4.1.2 Football Environment

Football environment (Kurach et al., 2020) developed by Google to test the agents in either single-agent or multi-agent setting. This is also done to ensure that it matches with the motivation example discussed in the introduction. We chose these three different scenarios to test the proposed approach.

1. 3_vs_1_with_keeper

2. Run-to_score_with_keeper

3. Run_pass_and_shoot_with_keeper

The observation space considered here is 113-dimensional vector as discussed in Kurach et al. (2020), full action space. To introduce stochasticity, two scenarios are considered:

**Case 1**: The idea is to replace the action suggested by the policy with any action from the action set with probability $\beta$.

**Case 2**: Here, we considered a more realistic case, where the actions are replaced only with "Similar actions" as per the noise model suggested in (2). For example, any policy which recommends action contains 'top' has to be replaced with set containing of the 'top', 'top-left' and top-right' actions only. Realistically, when attempting to kick top, the injured player won't be so naive to kick the ball behind him, i.e towards his own goal, but the impaired state might lead him to kick in any of these three directions with equal probability. Similarly, action containing 'left' has to be replaced with set containing either 'left' action or 'right' action. The subset of actions for every action is shown in Table 1.

To quantify the results, we used two metrics (i) % success rate and (ii) goal difference. In both cases, we train the models until they converged. The models are tested for 64 episodes with 100 different seeds.

### 4.1.3 IbForaging Environment

This environment contains agents and food resources randomly placed in a grid world. The agents navigate in the grid world and collect food resources by cooperating with other agents. The environment was modified by adding a couple of changes as mentioned herein.

100 food resources were placed in the $30 \times 30$ grid and 6 agents were chose to consume those resources. The episode is terminated either when there are no food resources available in the grid or the number of time steps reached $500$. Here out of 6 agents, three agents 3 are deterministic (1,3,4) and 3 are degraded (2,5,6) with degree of noise of $0.2, 0.4$, and $0.6$ respectively. For this problem context, both individual targets and global targets are considered. The targets were chosen for individual agents such that the sum of the targets exactly equals the total resource in the grid. The agents need to consume all the resources in the grid ( global target) while fulfilling their individual goals. The Global reward at any time step is measured by the sum of the difference between individual agent values and their targets

$$\text{Global\_Reward} = \sum_{i=1}^{6} (c_i - t_i) \tag{9}$$

where $c_i$ is the food consumed by the agent $i$ and $t_i$ is the target for the agent $i$.

**Modifications to IbForgaing environment**

The modifications to the environment are as follows.

1. We added two additional actions '**Carry on**' and '**Leave**' to the agents. The '**Carry on**' action enables the agent to store the food resources which they can subsequently leave in another time step and/or state for consumption of another agent. The '**Leave**' action enables to agent to drop the resources which they consumed.

2. Each agent is given a target for the amount of food resources they need to consume. For this, we modify the reward function by adding targets to them. Our eventual goal is to ensure all agents reach their targets. This means if some of the agents acted greedily beyond their target, they must yield and give up the extra resources for benefit of other agents.

These modifications are important as it enables us to measure the individual agent-wise contribution, i.e the reward attribution using them.

### 4.1.4 Water world Environment

The environment contains different pursuers, food resources, and poisons. The agents control the pursuers in order to consume as many food resources while avoiding the poison. The environment here is a continuous 2D space and each purser can move only within this space. The actions are also continuous and they represent the amount of thrust being added to both horizontal and vertical directions. Each pursuer has a number of evenly spaced sensors that can read the speed and direction of objects near the pursuer. This information is reported in the observation space and can be used to navigate the pursuers through the environment.

We considered 5 pursuers here with 10 poisons and 5 food resources. The observation space here is a $242$ element vector for each of the agents. The action space is a two-element vector that represents the thrust being added to the existing velocities in horizontal and vertical directions.

For this environment, we performed two sets of experiments with different degradation levels. The degradation value of agents are given in Table 2. Also, in each of these experiments, we considered two scenarios where the noise is added to both directions or either one of the two directions.

**Scenario 1**: In this scenario, we add the noise as per the model given in (3) for both the elements in the action vector. This may not be realistic as the probability of both motors failing at the same time is very low.

Table 2: Degradation values of agents used in the different experiments in the Waterworld environment

| Exp. No. | No. of noisy agents out of 5 agents | Degradation factor |
|---|---|---|
| 1 | 3 | [0.1, 0, 0.6, 0, 0.8] |
| 2 | 2 | [0, 0.4, 0, 0.9, 0] |

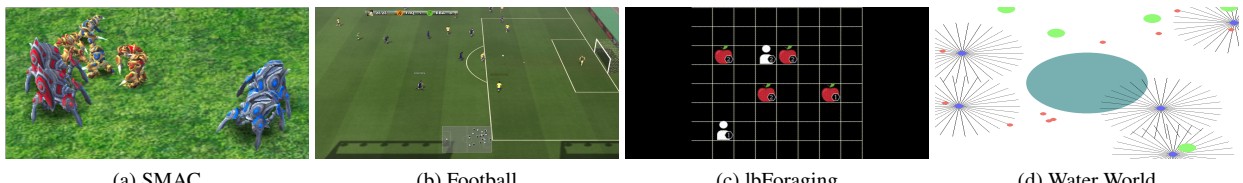

(a) SMAC       (b) Football       (c) lbForaging       (d) Water World

Figure 4: Snapshots of the environments used in this paper

**Scenario 2**: In this scenario, we add the noise as per the model given in (3) to either of the elements in the action vector. At every time step, we select randomly one of the two elements in the vector. Further, we add noise to the selected element.

A snapshot of all four environments is shown in Figure 4.

**Benchmark algorithms to compare**: To demonstrate the efficacy of the proposed approach in discrete action spaces, we chosen to compare the results with two baselines (i) QMiX (Rashid et al., 2018) and (ii) IQL (Tan, 1993). Also, in the case of continuous action space, the results are compared with FACMAC approach (Peng et al., 2021), as it is considered one of the benchmark method in continuous action space and logically derived from principles of QMIX. The main contribution of the work uses the existing baseline with only discount factor estimation on top, we can compare with the respective baselines instead of advanced ones. With the advanced ones, we expect similar improved results when we use them as underlying collaborative training algorithms. The reason is that advanced algorithms like MAVEN (Mahajan et al., 2019) attempt at improving global collaboration by use of non-monotonic functions or hastening the learning process by learning from each other's observations (Liu & Tan, 2022) but none deals with optimizing collaboration in a mix of noisy and accordant agents.

To quantify the performance of the proposed and existing approaches, we used multi-episodic evaluation along with multiple-seed approach as detailed in subsequent sections.

## 4.2 Results on SMAC Environment

In this section, experiments were performed on the above two environments with proposed DSDF, iterative penalization, QMIX and IQL. We skipped comparison with (Xu et al., 2018) as our proposed method concerns collaboration among multiple agents and optimizing global reward, rather than optimizing rewards of a single agent, which is a slightly different problem. Also value of $\gamma$ is same over entire episode which makes it restrictive. The pymarl library was utilized for the same (Samvelyan et al., 2019a).

### 4.2.1 Scenario 1

To evaluate performance of the agents, the training was paused after every 100 episodes and tested for 20 episodes. The plot of the % test winning episodes for the experiment index 1 (as per Figure 3a) is shown in the Figure 5a. Notably, the learning representation of discount factor which is the key part of proposed DSDF approach is trained only on the experiment 1 for each of the three map scenarios. For remaining 9 experiments, the respective trained learning representation was utilized to provide discount factor $\gamma_i$ value for each agent $i$.

From the plot it can be seen that the IQL performed poorly when two of the agents are noisy. Although QMIX gave good performance compared to IQL, it also settled at around $40\%$ winning rate. Both these methods performed below expectation as noisy agents look much into future with the choice of highest discount factor ($\gamma = 0.92$) which leads to poor planning and reduced coordination efficacy.

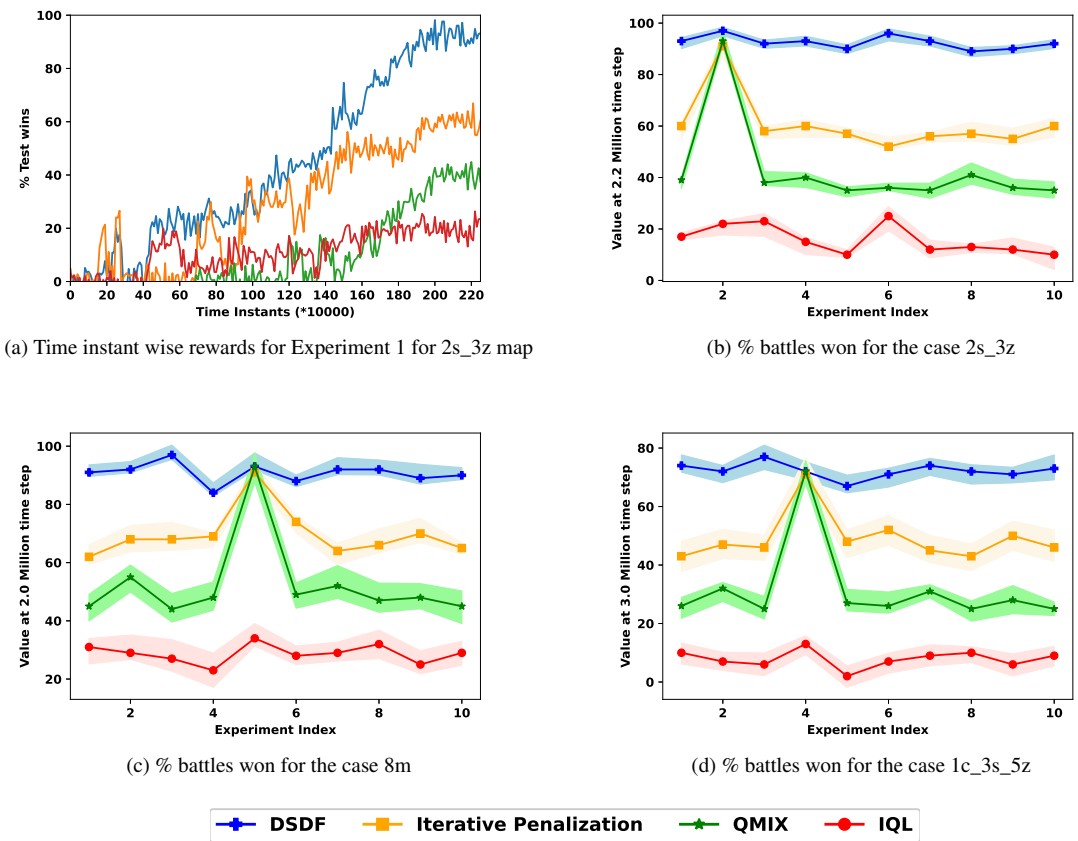

(a) Time instant wise rewards for Experiment 1 for 2s_3z map

(b) % battles won for the case 2s_3z

(c) % battles won for the case 8m

(d) % battles won for the case 1c_3s_5z

Figure 5: Returns obtained for SMAC environment for the experiments across 4 seeds in different map scenarios with confidence intervals. The dots in the figures 5b, 5c and 5d are connected for intuitive comparison. The proposed DSDF approach outperforms the existing methods and iterative penalization technique.

The iterative penalization method further improves the collaboration to $60\%$, while the proposed DSDF method exceeds others by achieving the top value of $95\%$. The reason being the discount factors are predicted by a non-linear network which utilizes the interactions among agents to decipher the mutual limitations and thus obtain appropriate discount values maximizing the global good.

For each experiment in all the three map scenarios, the proposed DSDF was evaluated along with the existing methods. The plot of the % test wins (obtained at the time instant given next to the map names) obtained for each experiment in respective map scenarios (i) 2s_3z ($220 \times 10^4$), (ii) 8m ($200 \times 10^4$) and (iii) 1c_3s_5z ($300 \times 10^4$) is shown in Figures 5b, 5c and 5d respectively. From the plots it is evident that the proposed DSDF approach outperforms all existing methods, which shows the efficacy of the proposed approach. It is also evident that the proposed DSDF approach performs better even when composition of noisy agents comprises about 75% of total number. The proposed approach also performs well for the case where all the agents are deterministic (experiment 2 for 2s_3z, experiment 5 for 8m and experiment 4 for 1c_3s_5z), which shows the generalization of the approach. The reason being the proposed DSDF approach dynamically chooses the best discount factor for each agent based on the agent's degradation (current capability) whereas the existing methods use constant discount factor.

### 4.2.2 Scenario 2

In this scenario, the agents continuously and monotonically degrade at varying rates. The experiment is started when all agents are accordant and learning representation along with the joint policy is trained from scratch. Now, whenever

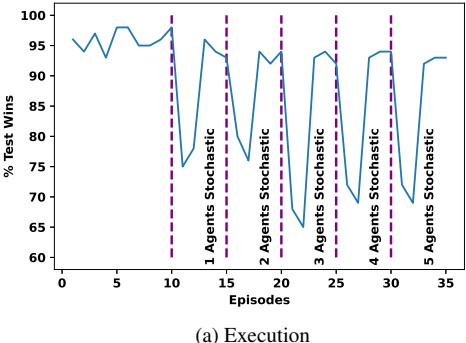
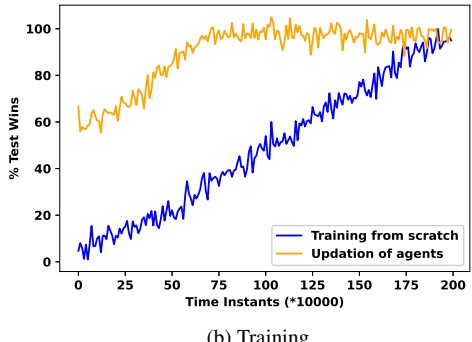

(a) Execution            (b) Training

Figure 6: Returns obtained using DSDF approach on SMAC environment with 8m map for continuous degradation. In first plot, the change in joint performance is depicted as each agent turns noisy and is subsequently retrained. In the second plot, we show a comparison of training performance when agents are trained from scratch as to when updated from their previous network values.

the collaborative performance degrades beyond a threshold due to change in agents' degradation, only the joint policy is updated from the previous values.

The performance of the agents during execution is shown in Figure 6a. During the $10^{th}$ episode, one of the agents turns noisy with a stochasticity of $0.4$. The agents' network is updated only when the execution performance falls below the threshold for consecutive two episodes. In this case, the threshold of degradation is chosen as $10\%$. Post retraining of agents' network, as seen in Figure 6a, the performance of the agents improve significantly. It should be emphasized that the learning representation of the discount factor network need not be updated. The re-training complexity is shown in Figure 6b in terms of number of time steps taken for convergence. The performance of the agents when the trained from scratch also is shown in Figure 6b. From the plot it is evident that retraining the agents' joint policy using transfer learning approach took significantly lesser time to converge than the traditional training from scratch. Hence the transfer learning based approach makes it increasingly feasible to apply the method in Industry 4.0 environments.

Returning to the main discussion, for every 5 episodes one accordant agent becomes noisy and noise factor changes for pre-existing noisy agents. The execution performance for $8m$ map scenario is shown in Figure 6a. From the plot, it is evident that the updation of agents' network improved the performance with only limited samples of re-training. Additionally, for other two map scenarios $2s\_3z$ and $1c\_3s\_5z$, the execution performance of the agents for the case of continuous degradation is shown in Figures 7a and 7b. From the plots, it is evident that the proposed method resulted in improved global performance even though more than half of the agents are noisy. Hence it may be concluded that the proposed approach can help in achieving desired global performance with minimal computation when an new agent turn noisy and/or noisy level of agent changes.

## 4.3   Results on the football environment

To demonstrate the efficacy of the proposed approach to handle stochastic agents, additional experiments were performed on the Google research football environment (Kurach et al., 2020). As explained at the start of this section, we performed experiments on this environment for two cases (i) replacing with entire action space and (ii) replacing only with subset of action space.

In these two cases, we considered 50% of agents to be noisy with fixed degradation value. The number of agents and their noise level are given in Table 3.

### 4.3.1   Case 1

For each scenario, the agents are trained until they converge. The training is paused after every 100 episode, tested for 32 episodes and the accuracy is tabulated. The plot of test accuracy obtained for scenario 3_vs_1_with_keeper is

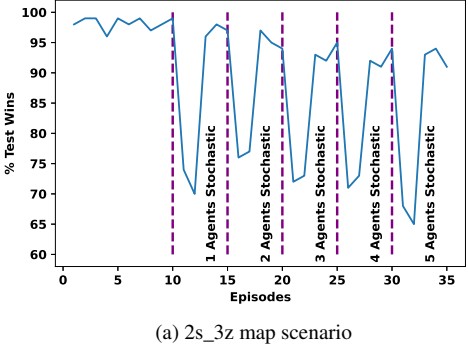

(a) 2s_3z map scenario

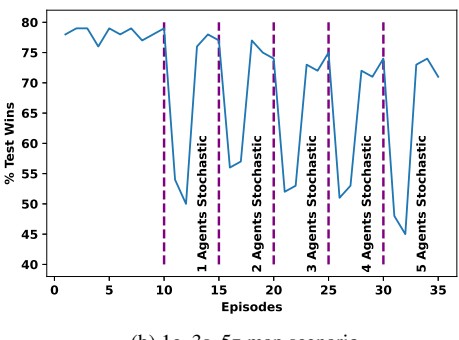

(b) 1c_3s_5z map scenario

Figure 7: Results obtained using DSDF approach on the SMAC environment for 2s_3z and 1c_3s_5z maps for the case of continuous degradation case. As expected, the proposed approach able to achieve a good performance even when an agent by agent turns noisy. In this way it can be ensured the proposed approach can achieve good collaboration even when some of the agents/all agents turn noisy.

Table 3: Number of agents and their noisy levels in each scenario in football environment

| Scenario | No. of agents | Noisy Level |
|---|---|---|
| 3_vs_1_with_keeper | 3 | $0, 0.2, 0.4$ |
| Run-to_score_with_keeper | 5 | $0.1, 0.3, 0.6, 0, 0$ |
| Run_pass_and_shoot_with_keeper | 6 | $0, 0, 0.6, 0.4, 0.7, 0$ |

shown in Figure 8a. From the plot, it is evident that the proposed approach consistently resulted in increased accuracy when compared with existing methods and also the iterative penalization method. We skipped the plot for two other scenarios as similar observations are obtained from them.

The final averaged accuracy along with errors obtained for all scenarios are shown in the Table 4. We also included the comparison with other benchmark methods in the same table. Also, we tabulated the averaged goal difference values between the teams along with their respective errors in Table 5. From both tables, it is evident that the proposed approach resulted in higher accuracy along with higher goal difference when compared with existing methods.

Table 4: % winning rate for Case 1 for different scenarios in football environment

| Scenario | Proposed Approach | Iterative Penalization | QMiX | IQL |
|---|---|---|---|---|
| 3_vs_1_with_keeper | $0.89 \pm 0.021$ | $0.46 \pm 0.034$ | $0.24 \pm 0.04$ | $0.06 \pm 0.02$ |
| Run-to_score_with_keeper | $0.84 \pm 0.034$ | $0.57 \pm 0.026$ | $0.27 \pm 0.041$ | $0 \pm 0.056$ |
| Run_pass_and_shoot_with_keeper | $0.78 \pm 0.078$ | $0.34 \pm 0.051$ | $0.24 \pm 0.14$ | $0.02 \pm 0.51$ |

### 4.3.2 Case 2

For case 2, the testing accuracy of the agents during the middle of training for scenario 3_vs_1_with_keeper is shown in Figure 8b. From the plot, it can be seen that the proposed approach consistently resulted in a higher success rate when compared with existing methods and also the iterative penalization technique. Also, it is observed that the success rate obtained here is higher than that of the previous case. The reason for this is explained below:

In case 1, the action suggested by the policy is replaced with any of the actions in the space. However, this is not correct as 'top' action can be replaced by an entirely different action 'down', which can result in a poor reward. On the other hand, in case 2, the 'top' action is replaced with either 'Top' or 'Top left' or 'Top right' actions and this can result in lessening the impact to global coordination than that of case 1. However, it should be noted that the stochastic behavior of the degraded agents still impacts the global coordination than following the actions suggested by policy (as is done by accordant agents).

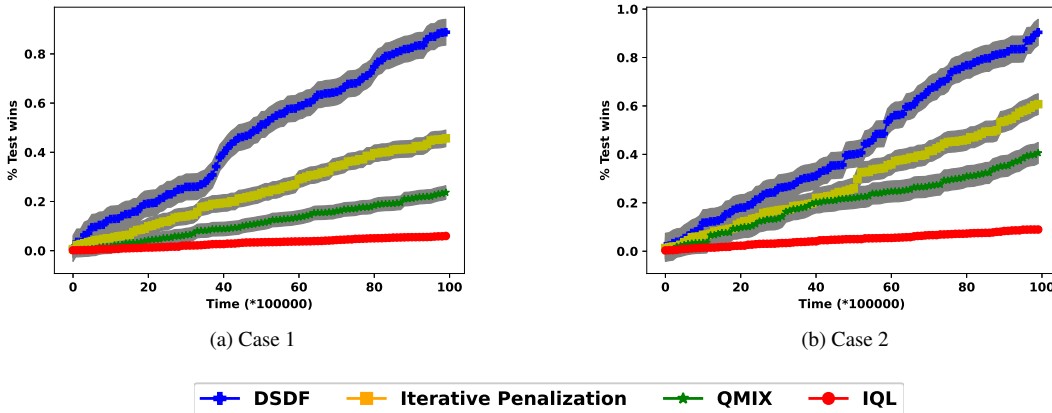

(a) Case 1            (b) Case 2

Figure 8: Results obtained across 6 seeds for both the cases when tested it on Scenario (3_vs_1_with_keeper. In both cases, it can be seen that the proposed approach resulted in a good percentage of test wins even when all the agents are noisy.

Table 5: Goal Difference for Case 1 for different scenarios in football environment

| Scenario | Proposed Approach | Iterative Penalization | QMiX | IQL |
|---|---|---|---|---|
| 3_vs_1_with_keeper | $1.47 \pm 0.031$ | $1.01 \pm 0.018$ | $-0.14 \pm 0.26$ | $-0.61 \pm 0.79$ |
| Run-to_score_with_keeper | $1.27 \pm 0.18$ | $0.29 \pm 0.027$ | $0.07 \pm 0.024$ | $0.01 \pm 0.089$ |
| Run_pass_and_shoot_with_keeper | $1.87 \pm 0.152$ | $0.85 \pm 0.124$ | $0.14 \pm 0.47$ | $-0.21 \pm 0.057$ |

Returning to the discussion, the final converged accuracy along with errors for each scenario is shown in Table 6. From the results, it is evident that the proposed approach results in higher accuracy when compared with the iterative penalization method and also the existing methods. Also, the accuracy obtained is higher than that of Case 1 owing to the reason explained above. The goal difference results between teams are shown in Table 7. From both the tables, it is evident that the proposed approach resulted in higher success rate and higher goal difference when compared with existing methods.

Hence in both cases, it can be seen that the proposed method results in significantly improved global coordination in the scenarios where noisy agents interact with accordant agents.

## 4.4 Results on lbForaging environment

In this experiment, all the 6 agents in the environment are assigned the target of 20. The sum of food resources available in the grid was restricted to a number $T$ such that the total sum of targets for individual agents is equal to $T$. **An important point to be noted is that the value of $T$ is not known to any agent during training or in execution.** Here the sum of targets is chosen to be $T = 120$. The setting necessitated the agents to collaborate within themselves to reach their respective targets.

The predictions of the discount factor values using DSDF method for all the time steps is shown in the Figure 9a. As seen in the figure, the agents' discount factor changes whenever the discount factor network is updated. From the plot,

Table 6: % winning rate for Case 2 for different scenarios in football environment

| Scenario | Proposed Approach | Iterative Penalization | QMiX | IQL |
|---|---|---|---|---|
| 3_vs_1_with_keeper | $0.92 \pm 0.02$ | $0.62 \pm 0.054$ | $0.41 \pm 0.09$ | $0.09 \pm 0.02$ |
| Run-to_score_with_keeper | $0.9 \pm 0.09$ | $0.71 \pm 0.031$ | $0.39 \pm 0.12$ | $0.07 \pm 0.15$ |
| Run_pass_and_shoot_with_keeper | $0.81 \pm 0.075$ | $0.47 \pm 0.142$ | $0.37 \pm 0.09$ | $0.12 \pm 0.123$ |

Table 7: % winning rate for Case 2 for different scenarios in football environment

| Scenario | Proposed Approach | Iterative Penalization | QMiX | IQL |
|---|---|---|---|---|
| 3_vs_1_with_keeper | $2.01 \pm 0.15$ | $1.57 \pm 0.09$ | $0.68 \pm 0.18$ | $0.31 \pm 0.21$ |
| Run-to_score_with_keeper | $1.75 \pm 0.14$ | $1.27 \pm 0.22$ | $0.84 \pm 0.042$ | $0.21 \pm 0.014$ |
| Run_pass_and_shoot_with_keeper | $1.92 \pm 0.24$ | $1.49 \pm 0.17$ | $0.48 \pm 0.13$ | $0.19 \pm 0.12$ |

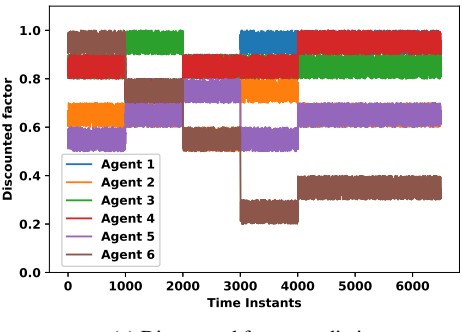

(a) Discounted factor prediction

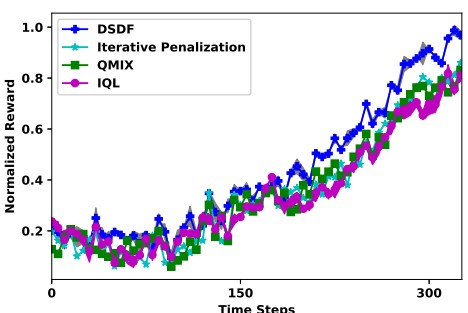

(b) Normalized global returns for correct resources

Figure 9: Returns obtained for lbForaging environment across 3 seeds along with discount factor prediction plot. The advantage obtained using DSDF approach when compared with existing methods is not significant here since the environment requires lesser collaboration between agents and is less complex in the number of states when compared with SMAC environments. However, the results for this environment is depicted as credit assignment for individual agents as evident here.

it can be observed that the values got almost saturated after some updates which shows the hypernetwork is converging and hence after this time the discount factor hyper network need not be updated. It should be noted the discount factor values will change (depending on the local observations and global state) with every batch of samples.

Also, it can be observed the discount factor values for accordant agents are higher ($>= 0.9$) which suggests the algorithm makes the respective utility functions depend more on the future values. This can be explained given the accordant agents need to look more into the future and decide on current actions. On the other hand, noisy agents should choose a lower discount factor so that they can plan short-term. From Figure 9a, it can be observed that the agents with a high degree of degradation i.e. they have less probability of executing actions given by the policy, use less discount factor. This is in accordance with the assumption that the more the noise in the agents, the less the discount factor value should be and vice-versa.

The performance of the individual agents are shown in Figure 10. In all the plots, the first 10 indices correspond to the latest 10 training episodes and the next indices correspond to 10 execution episodes. From the plots, it can be concluded that accordant agents, trained using the DSDF method exceeded their targets in most of the execution episodes. This was needed to achieve the desired global reward as the noisy agents fell short of their targets. On the other hand, the accordant agents trained with vanilla QMIX and IQL reached their targets in a few execution episodes and did not attempt to compensate for the performance of noisy agents. Notably, even the performance of the noisy agents trained using the proposed DSDF method as well as the iterative penalization method outperforms the vanilla QMIX and IQL. All of these could be attributed to the fact that the DSDF learning representation helps accordant agents realize the limitation of noisy agents and thereby assume additional workload to compensate for the degraded performance of noisy agents. Simultaneously as noisy agents attempt only the nearby goals reducing their total uncertainty over the trajectory, the global coordination improves.

The mean reward obtained for every time step with 95% confidence interval during evaluation is shown in Figure 9b. From the plot, it is evident that the proposed DSDF method resulted in a higher average reward when compared to other state-of-art methods. However, the state space has lesser complexity than the SMAC environment and hence the difference in global rewards is not as significant as in SMAC.

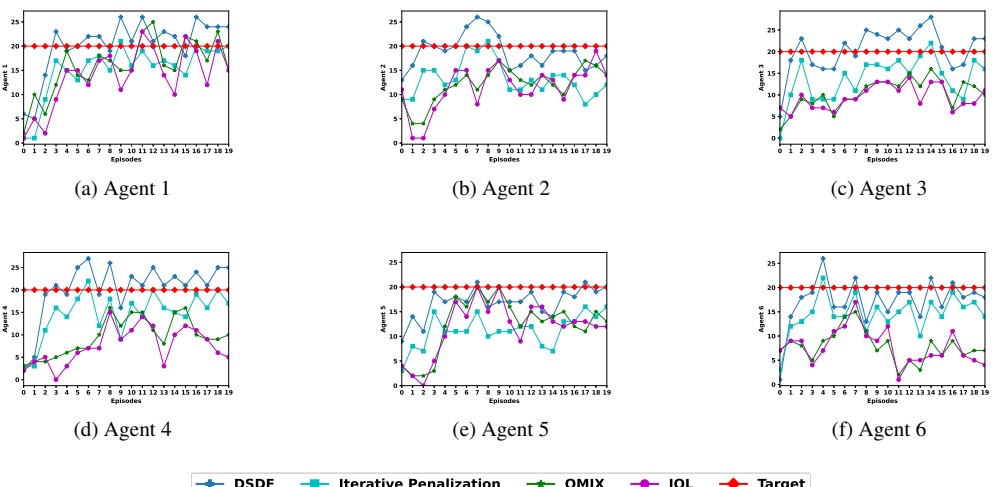

|      |      |      |
| :--: | :--: | :--: |
| (a) Agent 1 | (b) Agent 2 | (c) Agent 3 |
| (d) Agent 4 | (e) Agent 5 | (f) Agent 6 |

Figure 10: Comparison of individual agents performance using proposed DSDF approach vs existing approaches for lb-Foraging environment. One can observe the deterministic agents (1,3,4) achieved targets in almost all the test episodes whereas the noisy agents (2,5,6) stayed less than the targets. This confirmed our look-ahead strategy works better and results in good collaboration. The agents adjusted themselves owing to the nature of global reward formulation as per (1). Hence it is evident that the proposed approach resulted in deterministic agents taking a long-sighted approach (accumulating more resources) while noisy agents are short-sighted (accumulating exact or limited resources).

So in summary, based on individual reward attribution figure 10 the DSDF method helps in improving the performance of both accordant and noisy agents and thus outperforms the existing state of art methods.

## 4.5   Results on Water world environment

As per the experiment settings provided in table 2, the results are discussed here. In both experiments, $\epsilon^i$ is sampled from a Gaussian distribution $\epsilon^i \sim \mathcal{N}(0, 0.0001) \ \forall \ i$

### 4.5.1   Experiment 1

In this experiment, we chosen 3 out of 5 agents as noisy. The degradation values for each of the agents are given in the first row of Table 2. **For this experiment the agents are trained using the DSDF approach along with the training of discount factor hypernetwork.** We stopped the training for every 100 time-steps and evaluate the trained model on 2 episodes. We repeat the training until the model converges.

The plot of the average returns obtained for the scenario 1 is shown in Figure 11a, where we add noise to both the elements in the action vector. From the plot, it can be seen that the proposed DSDF approach achieved a higher average return when compared with FACMAC approach (which is based on policy gradient). Also, for the scenario 2 where we add noise to only either of the element in the action vector, the comparison plot is shown in Figure 11b. Here also it is evident that the proposed DSDF outperforms the FACMAC baseline.

### 4.5.2   Experiment 2

In this experiment, 2 out of 5 agents are chosen to be noisy as per the degradation values given in Table 2. **For this case, the trained discount factor network from the previous experiment is used. We only train policies for the agents.** The same process of performance evaluation is followed as in the previous experiment.

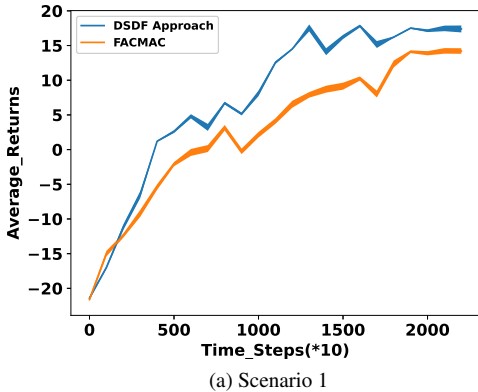
(a) Scenario 1

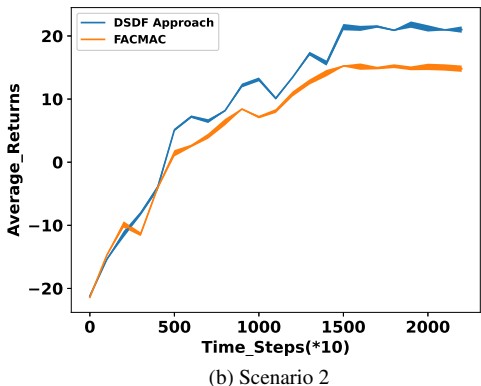
(b) Scenario 2

Figure 11: Comparison of the proposed DSDF with FACMAC approach on the Water World environment on experiment 1 across 2 seeds. For both scenarios, it can be seen the proposed DSDF approach outperforms the FACMAC approach.

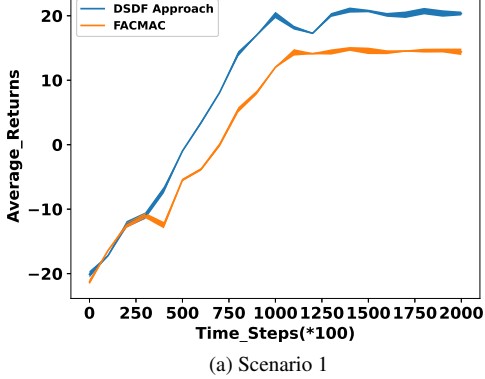
(a) Scenario 1

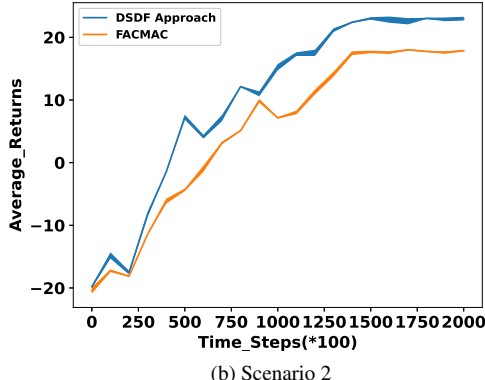
(b) Scenario 2

Figure 12: Comparison of the proposed DSDF with FACMAC approach on the Water World environment on experiment 2 across 2 seeds. For both scenarios, it can be seen the proposed DSDF approach outperforms the FACMAC approach.

The plot of the average returns for the scenario 1 where we add noise to both the components is shown in Figure 12a. From the plot, it can be seen that the proposed DSDF approach achieves higher average returns when compared with the FACMAC approach. Also, for scenario 2 the plot of average returns obtained using the DSDF and FACMAC approach is shown in Figure 12b. From the plot, it is evident that the proposed approach also achieves higher average returns when compared to FACMAC approach.

Hence from both experiments it can be concluded that the proposed DSDF approach can achieve superior collaboration in environments with continuous action space even when some of the agents are noisy.

## 5 Conclusion

In this paper, we propose a novel method DSDF, which addresses a mix of accordant and noisy agents and learns a collaborative joint policy. Our proposed method can be combined with any state of art MARL algorithms without much impact to existing computational complexity. Two approaches to learning the joint policy have been outlined: (i) training agent policy from scratch and (ii) from the previous state of degradation, for the case when noise levels

change continuously and monotonically. In both cases, the learning representation to compute the discount factors is learned only once and re-used for subsequent states of degradation.

Our proposed method is tested on the four different environments, SMAC, Football, lbForaging, and Water World, and demonstrated clear improvement in results, in both discrete and continuous action spaces, when compared with benchmark methods. To the best of our knowledge, this is the first time, such an interaction involving degraded/noisy and accordant agents has been explored in the context of multi-agent reinforcement learning. Future directions include extending the technique to more generalized situations where the environment might also be noisy in regions.

The method can have significant usage in reducing maintenance costs for future Industry 4.0 scenarios or robots in space exploration, where multiple robots might need to coordinate for a global objective.

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

# Appendix

## A  Combined loss function used in the work

The general loss function of QMIX is

$$\mathcal{L}(\theta) = \sum_{t=1}^{B} (y_t - Q_{tot}(\tau, \mathbf{a}, s : \theta) \tag{10}$$

where $\theta = [\theta_i \ \forall \ i = 1, \cdots, N \ , \theta_{tot}]$ is the set of parameters of $N$ utility agents and mixing network. Now, if we expand $y_t$

$$y_t = r_t + \gamma \ \max_{a'} \ Q_{tot}(\tau', \mathbf{a}', s' : \theta^-) \tag{11}$$

Now, instead of using a single $\gamma$ value, we will take the $\gamma$ inside the equation to handle the stochastic agents

$$y_t = r_t + \max_{a'} \ g(\mathbf{a}', s', \gamma_i Q^i, \theta_{tot}^-) \tag{12}$$

with $i$ ranges from $1, \cdots, N$. Here $g(.)$ is the target network of mixing network architecture which is parametrized by $\theta_{tot}^-$, $Q^i$ is the individual utility function of agent $i$. The individual utility function can be represented as

$$Q_t^i = f^i(o_t^i, a_{t-1}^i, \theta_i^-) \tag{13}$$

where $f^i$ is the function of the $i^{\text{th}}$ utility network paramterized by $\theta^i$.

Substituting (13) in (12) gives

$$y_t = r_t + \max_{a'} \ g(\mathbf{a}', s', \gamma_i f^i(o_t^i, a_{t-1}^i, \theta^{i^-}), \theta_{tot-}) \tag{14}$$

In the paper, the idea is to predict the $\gamma^i \ \forall \ i = 1, \cdots, N$ and use them to scale the predicted Q-values of individual agents from the network. Now, we will replace the $\gamma^i$ with output of the network $\theta_\gamma$. The replaced equation is

$$
\begin{aligned}
y_t &= r_t + \max_{a'} \ g(\mathbf{a}', s', f_\gamma(o_t^1, \cdots, o_t^N, \theta_\gamma) \\
&\quad f^i(o_t^i, a_{t-1}^i, \theta^{i^-}), \theta_{tot}^-) \\
&= r_t + \max_{a'} \ g(\mathbf{a}', s', f_\gamma(o_t^1, \cdots, o_t^N, f_h(\theta_h, s')) \\
&\quad f^i(o_t^i, a_{t-1}^i, \theta^{i^-}), \theta_{tot-})
\end{aligned}
\tag{15}
$$

where $f_\gamma$ is the discounted factor hyper network which is parametrized by $\theta_\gamma$ and $f_h$ is the hypernetwork function which is parametrized by $\theta_h$. The hypernetwork $\theta_h$ is

Replacing the value of $y_{tot}^i$ from (7) with that of (4) we obtain the loss function as

$$\mathcal{L}(\theta, \theta_h) = \sum_{t=1}^{B} \left( r_t + \max_{a'} \ g(\mathbf{a}', s', f_\gamma(o_t^1, \cdots, o_t^N, f_h(\theta_h, s')) \right)$$
$$f^i(o_t^i, a_{t-1}^i, \theta^{i^-}), \theta_{tot^-}) - Q_{tot}(\tau, \mathbf{a}, s : \theta) \tag{16}$$

Replacing the agent utility networks function (13) in the (16) gives

$$\mathcal{L}(\theta, \theta_h) = \sum_{t=1}^{B} \left( r_t + \max_{a'} g(\mathbf{a}', s', f_\gamma(o_t^1, \cdots, o_t^N, f_h(\theta_h, s')) \right)$$
$$f^i(o_t^i, a_{t-1}^i, \theta^i), \theta_{tot}^-) - Q_{tot}(\tau, \mathbf{a}, s : \theta) \tag{17}$$

In QMIX, the $\theta_{tot}$ is computed by a separate hypernetwork which will update the weights from a hypernetwork.

In this work, we use the combined loss function in (17) to estimate the $\theta$ and $\theta_h$.

## B  Details of the networks used in SMAC environment example

We used the exact QMIX network architectures for this environment as used in pymarl library (Rashid et al., 2018). In addition, we chose the $\boldsymbol{\theta}_h$ to be a 1 hidden layer fully connected network with an input layer followed by a layer of 64 nodes, followed by an output layer. The $\boldsymbol{\theta}_\gamma$ is a 1-layer fully connected network with 64 nodes.

The value of $P$ is chosen to $0.0001$ in the iterative penalization method.

## C  Details of the networks used in Football environment example

We used the exact QMIX network architectures for this environment as used in (Huang et al., 2021). In addition, we chose the $\boldsymbol{\theta}_h$ to be a 2-layered fully connected network with an input layer followed by the first layer of 64 nodes, the second layer of 32 nodes, and the output layer. The $\boldsymbol{\theta}_\gamma$ is a 1-layer fully connected network with 32 nodes.

The value of $P$ is chosen to $0.0001$ in the iterative penalization method.

## D  Details of the networks used in lbForaging environment example

Below are the details of the networks we used in the implementation example

---

**Implementation details used in the paper**

---

**Environment Parameters**

1. Number of agents - 6

2. Number of food resources - 100

3. Grid size - $30 \times 30$

4. Sight - 10

5. Max Episode steps - 1000

6. Cooperation - True

7. Max player level - 20

8. Stochastic level of agents - [0,0.2,0,0,0.3,0.6]

**Network Parameters**

1. Agents utility networks are 3-layer networks. First layer of the network is MLP with $318$ nodes followed by GRU layer with final layer being MLP layer with 8 nodes output.

2. Mixing network is chosen as two layer fully connected network with 6 inputs nodes in the first layer and eight output nodes in the last layer.

3. Mixing hypernetwork is also chosen as two layers with 40-node first layer followed by another layer matching mixing layer weights dimension.

4. Discount factor network is also chosen as two-layer network with $324$ nodes in the first node followed by $6$ output nodes in the last layer.

5. Discount factor hypernetwork is chosen as two layers which will match the dimension of the discounted factor network.

6. The learning rate for all the networks is chosen to be $0.96$ since at this learning rate we achieved good results.

7. Discount factor values for the QMIX and IQL methods are chosen as $0.92$. This value is obtained after a grid search.

The value of $P$ is chosen to $0.01$ in the iterative penalization method.

# E  Details of the networks used in Water World environment example

Below are the details of the networks we used in the implementation example

---

**Implementation details used in the paper**

---

**Environment Parameters**

We used similar environment parameters as given in [1].

**Network Parameters**

1. Agents utility networks are 3-layer networks. First layer of the network is MLP with $242$ nodes followed by GRU layer with final layer being MLP layer with 2 nodes output.

2. Mixing network is chosen as two-layer fully connected network with 5 input nodes in the first layer and one output node.

3. Discount factor network is also chosen as a two-layer network with $1220$ nodes in the first node followed by $5$ output nodes in the last layer.

4. Discount factor hypernetwork is chosen as two layers which will match the dimension of the discounted factor network.

5. The learning rate for all the networks is chosen to be $0.82$ since at this learning rate we achieved good results.

6. Discount factor value for the FACMAC method is chosen as $0.94$. This value is obtained after a grid search.

## F  Applicability to Industry 4.0 scenario

Reinforcement learning has many interesting applications in different areas. [2] lists out some important challenges of applying reinforcement learning in real world problems. One such problem is noisy actions where the agent takes different actions (due to environment noise or agent noise) than that of policy.

With respect to the applicability of the problem in real-life application, may we point out, that the same is quite pertinent to the Industry 4.0 scenario, where a certain agent in a factory floor could get degraded and as a result, the entire output of the floor would be affected till the degraded agent is changed.

Leveraging our proposed method, the degraded agents could be used whereby they can leverage the limited capability while the other agents can take up longer trajectory jobs and thereby compensate, in parts, for the degraded ones. Such a mechanism could reduce the frequency of maintenance and thereby the total downtime in a factory floor. Such applications of coordinated robotic manufacturing are expected to become widespread in next 5 years. As an example, please refer to [3], where the discussion refers to the frequency of maintenance in industrial robots, which needs to be done daily, every 600 hours, and every 5000 hours. We believe, our method has the potential to reduce a good part of these maintenance hours thereby saving cost. Additionally, the projected need and complexity of managing multiple synchronized robots on a factory floor is also discussed in Section 5 of [4].

Additionally with increased traction in space exploration, lot of such tasks might need coordinated robots. When one such robot degrades or has a partial malfunction, it cannot be readily replaced. Hence the team strategy needs to be about extracting the best out of the constrained situation and algorithms like DSDF could provide a feasible solution in such circumstances.

[1] https://pettingzoo.farama.org/environments/sisl/waterworld/

[2] Dulac-Arnold, Gabriel, Daniel Mankowitz, and Todd Hester. "Challenges of real-world reinforcement learning." arXiv preprint arXiv:1904.12901 (2019).

[3] https://www.sdcautomation.com/blog/preventative-maintenance-for-industrial-robots/

[4] Marvel, Jeremy & Bostelman, Roger & Falco, Joseph. (2018). Multi-Robot Assembly Strategies and Metrics. ACM Computing Surveys. 51. 1-32.

