# OpenReview forum: "DSDF: Coordinated look-ahead strategy in multi-agent reinforcement learning with noisy agents"
_TMLR — Rejected by TMLR_

### Review · Reviewer_Pdpm · 2023-02-21

**Summary Of Contributions:**

This work studies a practical cooperative MARL scenario, where some of the agents degrade and perform noisy actions due to practical reasons like wear and tear of machines. The noisy/degraded agents add non-trivial challenges to learning coordination especially when continual degradation occurs.

In the context of MARL with CTDE, the authors propose two solutions, i.e., Iterative Penalization and Deep Stochastic Discounted Factor (DSDF), with the latter one as the main point of this paper. The key idea of the two solutions is to adjust/learn discounted factors for individual agents according to the degree of degradation. With adaptive discounted factors, the degraded agents learn to behave by optimizing a generally myopic objective that pertains to its degraded ability; while the agents also learn to adjust their policies to coordinate with the degraded agents.

The proposed methods are incorporated into QMIX and evaluated in several environments in SMAC, Google Football and ibForaging. With a series of scenarios with specific settings of agent degradation, the experiments demonstrate the effectiveness of DSDF under both fixed degradation and continual degradation.


**Audience:**

Yes

**Claims And Evidence:**

No

**Requested Changes:**

- I recommend the authors to polish and improve the presentation, especially the notations. A few concrete places are mentioned in Strengths and Weaknesses part.
- I recommend the authors to add more clarification, explanation and justification on the proposed method and the experiments according to my detailed comments in Strengths and Weaknesses part.


**Strengths And Weaknesses:**

Strengths:
+ I appreciate the authors’ efforts in studying the degraded/noisy agents in cooperative MARL, although simple settings and preliminary attempts are made in this work. It is a practical problem to consider in real-world scenarios and is of interests to RL practitioners.
+ The proposed methods are evaluated in three types of popular environments with both fixed degradation and continual degradation settings.


&nbsp;

Weaknesses (Concerns & Questions):
- Although I can understand most content of this work, the paper is poor in writing, especially the symbols and notations are inconsistent and not self-contained at some places. To list a few:
	- The notation of action is $a$ before page 5 and becomes $u$ after.
	- The usage of bold notations is inconsistent in this paper.
	- For the observation function, is it supposed to be a mapping from state and agent index to observation, rather than from state and action?
	- In Equation 3, I recommend using the subscript rather than the superscript.
	- According to Equation 5,6, I think the expression of $f_{\gamma}(.)$ in Equation 6 is incorrect since the output of $f_{\gamma}(.)$ is a vector of individual discounted factors.
	- There are duplicated sentences in Section 3.1.1.
	- I think using Scenario 1/2 rather than Approach 1/2 in Section 4.1.1 will help in eliminating the ambiguity.
- The presentation of the proposed method is not clear enough. Some important details are missing. I list some of these places below:
	- What are the concrete structures of the hypernetwork $\theta_{h}$ and its output $\theta_{\gamma}$?
	- In Section 3.1.1, the authors mention ‘use the average value of discounted factor during the execution phase’. It is confusing to me because the agents of a CTDE algorithm like QMIX, execute according to their individual $Q$ networks. I do not see how to use the discounted factor during the execution phase. The authors need to clarify this point.
	- In Algorithm 2 Line 16, how exactly to check the convergence of $\theta_{h}$?
- The soundness of the proposed method is questionable:
	- For Equation 6, I have a concern on discounting the individual Q values before inputting them into the value-mixing network. For QMIX, it is (nonlinear) monotonic network rather than a linear network (like used by VDN). Therefore, I am worried about the numerical correctness and rationality of this operation. I recommend the authors to provide some justifications on this point.
	- The detection of continual degradation by the performance degradation between two consecutive episodes (Algorithm 3 Line 4) is problematic in my opinion. I will agree this method is reliable only when the environment and the agents’ policies are deterministic.
	- Since the learned discounted factors depend only on the state and the observations, I do not see how it generalizes to different degrees of agent degradation. Concretely, it does not make sense to me that in Section 4.2, the learning representation of discount factor needs no more training for later experiments. I think the authors need to explain more on this point.
- For the experiments, some perspectives of the designed settings and the reported experimental results need more explanations:
	- How are the experiments shown in Figure 4 determined?
	- The results reported in Figure 6a and Figure 7 are somewhat surprising to me. Even with 5 degraded agents with a stochasticity of 0.4, QMIX also achieves high win rates. I cannot understand the results although I take into consideration the effect of DSDF.

---

> ### Author Response · Authors · 2023-03-15
> **Respones to comments given by Reviewer Pdpm (1/3)**
>
> **The notation of action is $a$ before page 5 and becomes $u$ after.**
>
> **The usage of bold notations is inconsistent in this paper.**
>
> We sincerely apologize for the notational issues. We revised all the
> notations and rechecked them in the revised paper.
>
> **For the observation function, is it supposed to be a mapping from
> state and agent index to observation, rather than from state and
> action?**
>
> Yes, the observation function should map from state to agent index. Due
> to a mismatch of notations, it is wrongly understood. We rectified it in
> the revised paper.
>
> **In Equation 3, I recommend using the subscript rather than the
> superscript.**
>
> As per the reviewer's suggestion we changed the superscript to script in
> the revised paper.
>
> **According to Equation 5,6, I think the expression of $f_\gamma(.)$ in
> Equation 6 is incorrect since the output of $f_\gamma(.)$ is a vector of
> individual discounted factors.**
>
> Yes, the output of the $f_\gamma(.)$ is vector of discounted factors. We
> modified the expression to reflect the vector notation in the revised
> version of the paper.
>
> **There are duplicated sentences in Section 3.1.1.**
>
> We removed the duplicated sentence in the revised paper.
>
> **I think using Scenario 1/2 rather than Approach 1/2 in Section 4.1.1
> will help in eliminating the ambiguity.**
>
> We changed the wording, from \"Scenario\" to \"Approach\" to eliminate
> ambiguity.
>
> **What are the concrete structures of the hypernetwork $\theta_h$ and
> its output $\theta_\gamma$?**
>
> In this work, both $\boldsymbol{\theta}_h$ and
> $\boldsymbol{\theta}_\gamma$ are fully connected networks. We chose the
> number of layers as per the environment we train the agents on. For
> example,
>
> -   In the case of the football environment, we chose the
>     $\boldsymbol{\theta}_h$ to be a 2-layered fully connected network
>     with $64$ and $32$ nodes. The $\boldsymbol{\theta}_\gamma$ is a
>     1-layer fully connected network with $32$ nodes.
>
> -   In the case of the SMAC environment, we chose the
>     $\boldsymbol{\theta}_h$ to be a 1-layer fully connected network with
>     $64$ nodes. The $\boldsymbol{\theta}_\gamma$ is a 1-layer fully
>     connected network with $64$ nodes.
>
> -   In the case of the lbForaging environment, we chose the
>     $\boldsymbol{\theta}_h$ to be a 1-layer fully connected network with
>     $32$ nodes. The $\boldsymbol{\theta}_\gamma$ is a 1-layer fully
>     connected network with $32$ nodes.
>
> We have added these details to the appendix (page 22) of the revised
> paper.
>
> **In Section 3.1.1, the authors mention 'use the average value of
> discounted factor during the execution phase'. It is confusing to me
> because the agents of a CTDE algorithm like QMIX, execute according to
> their individual networks. I do not see how to use the discounted factor
> during the execution phase. The authors need to clarify this point.**
>
> Thanks for pointing it out. We missed mentioning that this is in the
> training phase. The correct statement, thus, will be 'use the average
> value of discounted factor during the training phase' and it is
> corrected in the revised version of the paper on page $5$. In short,
> this is a summary of the iterative penalization method which is a
> three-step approach (i) penalize the discounted factor for every
> mismatch between the action decided by the policy and applied action,
> and arrive at averaged discounted factor, (ii) Use the averaged
> discounted factor to train the MARL agents and (iii) Use the trained
> agents during the execution phase.
>
> **In Algorithm 2 Line 16, how exactly to check the convergence of
> $\boldsymbol{\theta}_h$?**
>
> The condition is
> $||\boldsymbol{\theta}_h^{k+1}-\boldsymbol{\theta}_h^{k}||_2^2 \leq \epsilon$,
> where $\epsilon$ is a smaller value closer to $0$. It is updated in the
> revised paper.

---

> > ### Author Response · Authors · 2023-03-15
> > **Respones to comments given by Reviewer Pdpm (2/3)**
> >
> > **For Equation 6, I have a concern on discounting the individual Q
> > values before inputting them into the value-mixing network. For QMIX, it
> > is (nonlinear) monotonic network rather than a linear network (like used
> > by VDN). Therefore, I am worried about the numerical correctness and
> > rationality of this operation. I recommend the authors to provide some
> > justifications on this point.**
> >
> > To compare the two techniques, it should be noted that in the proposed
> > DSDF method, during training the estimated Q-values of the individual
> > agents are sent to the mixing network to arrive at the $Q_{tot}$ value.
> > This forms the predicted $Q_{tot}$ value (this step is the same in QMiX
> > and DSDF). Thereafter in DSDF, to arrive at the actual values, we scale
> > the individual agents' Q-values of the next states with the predicted
> > discount factor values (from the discount factor network) and add them
> > to the reward obtained from the environment. Whereas in vanilla QMiX,
> > the discount factor used is a pre-determined and fixed scalar value for
> > all the agents. In both approaches, the mixing network and individual
> > agents' Q-networks are trained based on the difference between the
> > predicted value and the actual value.
> >
> > As is evident the difference between the proposed approach and vanilla
> > Q-Mix is that we scale the individual Q-values of the next state with
> > dynamic discount factors (predicted using the discount factor network)
> > instead of the fixed discount factor. The discount factor network is
> > constrained such that the output is always a non-negative number between
> > $(0,1)$.
> >
> > The principle assumption of the QMiX method is to have a monotonicity of
> > $Q_{tot}$ w.r.t $Q^a$ of individual agents. Also, to ensure positive
> > $Q_{tot}$ it enforces a non-negativity constraint on the weights of the
> > mixing network. Given the discount factor network yields a number
> > between $(0,1)$, it does not interfere with any of the aforesaid
> > assumptions. So, we believe the proposed changes to QMiX will not
> > violate any critical assumptions. If advised, we can add this
> > explanation to the revised paper.
> >
> > **The detection of continual degradation by the performance degradation
> > between two consecutive episodes (Algorithm 3 Line 4) is problematic in
> > my opinion. I will agree this method is reliable only when the
> > environment and the agents' policies are deterministic.**
> >
> > In our case, the environment remains the same without any change or
> > drift. However, the agents slowly turn stochastic with varying degrees
> > of degradation. This noise or uncertainty is different from a stochastic
> > policy. If the basic agent policy is stochastic, without any external
> > noise, the average performance, over an episode, is not expected to
> > degrade. But given the degradation is substantial it can only be
> > attributed to the noise in agents. The degradation can be measured by a
> > predetermined factor, like win rate in SMAC, the total amount of food
> > consumed in lbForaging, and Goal difference in Google soccer. In the
> > real world, depending on the use case, it might need more than two
> > episodes to arrive at a conclusion.

---

> > > ### Author Response · Authors · 2023-03-15
> > > **Respones to comments given by Reviewer Pdpm (3/3)**
> > >
> > > **Since the learned discounted factors depend only on the state and the
> > > observations, I do not see how it generalizes to different degrees of
> > > agent degradation. Concretely, it does not make sense to me that in
> > > Section 4.2, the learning representation of the discount factor needs no
> > > more training for later experiments. I think the authors need to explain
> > > more on this point.**
> > >
> > > From our point of view, the discount factor network learns a
> > > relationship between the global state of the environment and also the
> > > local observations. Using the learned relation the network can able to
> > > predict the optimal discount factor values. Since the relationship
> > > depends on the degradation value of the agents (as we assume a similar
> > > environment across the entire training phase) the learned discounted
> > > factor network works even when degradation changes.
> > >
> > > For more insight into the interactive behavior of agents, we did
> > > additional experiments where we trained the discount factor network from
> > > scratch with changes in degradation values and compared it to the case
> > > where we have trained it once. We chose three different experiment
> > > indices (1,2,3) as shown in Figure 4a. These experiments correspond to
> > > map scenario $1c\_3s\_5z$ in the SMAC environment. We compared two
> > > scenarios here
> > >
> > > 1.  Train the discount factor network in Experiment $1$. Now, use the
> > >     trained discounted factor network to train the policies for
> > >     experiments $2$ and $3$.
> > >
> > > 2.  Train the discount factor network from scratch and also the policies
> > >     in each of experiments $2$ and $3$.
> > >
> > > The results are shown in the below table
> > >
> > > Table: Comparison of average % battles win for SMAC environment with map
> > >   $1c\_3s\_5z$ with discounted factor network trained on experiment $1$
> > >   vs trained from scratch for the experiment.
> > >
> > > | Experiment  Index | Pre-trained Network | Trained from  scratch |
> > > |:-----------------:|:-------------------:|:---------------------:|
> > > |         2         |    $75.7\pm0.02$    |      $76\pm0.02$      |
> > > |         3         |    $78.1\pm 0.06$   |     $78.1\pm0.01$     |
> > >
> > >
> > >
> > > From the table it can be seen that the usage of the \"trained once\"
> > > discount factor network results in similar performance when compared to
> > > that of trained from scratch. Hence, it is evident that the discount
> > > factor networks learn the relationship between the state of the
> > > environment and observations of agents and can be used to generalize the
> > > different degrees of degradation.
> > >
> > >
> > > **How are the experiments shown in Figure 4 determined?**
> > >
> > > We used a two-step approach to design the experiments. In the first
> > > step, we chose a random number of $N$ agents to be degraded. In the
> > > second step, we have chosen the degradation value for each agent as the
> > > random number between \[0.1,0.9\]. This forms one experiment for us
> > > which is represented as one row in the heatmap. Likewise, we ran 10 such
> > > experiments for each map scenario as shown in Figure 4.
> > >
> > > **The results reported in Figure 6a and Figure 7 are somewhat surprising
> > > to me. Even with 5 degraded agents with a stochasticity of 0.4, QMIX
> > > also achieves high win rates. I cannot understand the results although I
> > > take into consideration the effect of DSDF.**
> > >
> > > Figure 6a and 7 corresponds to the situation where DSDF is applied on
> > > top of QMiX framework to enhance the collaboration which in turn helps
> > > it to achieve high win rates. Here the proposed DSDF approach improves
> > > the averaged rewards significantly when a new agent becomes noisy. From
> > > figure 6b, it can be seen that the updation of policies of agents
> > > requires fewer samples to achieve good performance when compared to that
> > > of training from scratch. It should be noted we are not showing a
> > > comparison with QMiX here but just to show the effectiveness of our
> > > method in a realistic scenario of continuous degradation.

---

### Review · Reviewer_DwDm · 2023-02-23

**Summary Of Contributions:**

The paper proposes a method termed Deep Stochastic Discount Fact DSDFIn this paper, the authors propose a new method for Multi-Agent Reinforcement Learning that addresses the issue of degraded and noisy actions in some agents during the execution phase. Current methods aim to train agents towards a coordinated action pattern for optimal joint policy, but fail to provide effective coordination when some agents perform poorly. The authors claim to demonstrate how random noise in agents due to degradation or aging can contribute to unsatisfactory global rewards and add to the uncertainty in coordination. To address this issue, they propose the Deep Stochastic Discount Factor (DSDF) method which tunes the discount factor for each agent based on their degree of degradation. The DSDF approach is designed to handle changes in the degree of degradation over time, without extensive retraining.

**Audience:**

Yes

**Claims And Evidence:**

No

**Requested Changes:**

- Use angle brackets `\langle` for defining the Dec-POMDP tuple.
- Page 4: GLIE property Singh et al. (2000) should use `\citep`.
- Page 5: Calling $P$ a factor is slightly confusing as it seems to be subtracted rather than multiplied.
- More clarity on experiments

**Strengths And Weaknesses:**

**Strengths**

The paper explores an alternative viewpoint for coordination under non-ideal conditions compared to the difficult ad-hoc coordination problem. Non-stationarity of real world systems is well known and standard formulations often don't try to model this phenomenon and how this would effect coordination in a multi-agent system. The authors evaluate their proposed approach in multiple modified benchmark scenarios.

**Weaknesses**

There are several issues with the paper, particularly regarding the choice of evaluation scenarios in light of their motivations and current evaluation standards for reinforcement learning methods.

Given the motivations for Industry control applications and formulating the degradation problem as a random action problem, choosing domains with discrete actions only as the test environment seems unwise. While the authors try to come up with scenarios to justify their choices for "meaningful" randomness, like even an injured player is unlikely to kick the ball backward "randomly", continuous control actions would have been way better domains to test, for example [2].

Additionally it's unclear if the recommendation from [1] to _train_ with multiple seeds was followed. From the description in text it seems only multiple runs were done for evaluation which is not enough especially given the added randomness. Some discussion discussing the distinction from ad-hoc teaming approaches in MARL would have been relevant and possibly better comparison baselines.
Lastly, during online adaptation with continuous degradation, it is unclear if the paper still assumes access to global state information, and if so, whether it still aligns with the CTDE paradigm.

Considering the primary contribution is another network to train a scalar parameter, the method may introduce considerable complexity without clear benefits.


[1] https://arxiv.org/abs/1709.06560
[2] https://pettingzoo.farama.org/environments/sisl/

---

> ### Author Response · Authors · 2023-03-15
> **Respones to comments given by Reviewer DwDm (1/2)**
>
> **Given the motivations for Industry control applications and
> formulating the degradation problem as a random action problem, choosing
> domains with discrete actions only as the test environment seems unwise.
> While the authors try to come up with scenarios to justify their choices
> for \"meaningful\" randomness, like even an injured player is unlikely
> to kick the ball backward \"randomly\", continuous control actions would
> have been way better domains to test, for example \[2\].**
>
> We agree with the reviewer that the industrial scenario generally
> resembles an environment with continuous action space. However discrete
> action space algorithms have been implemented in real-world scenarios
> which are essentially continuous by effective binning of the action
> space [1]. Our work is a first in the direction of
> adjusting discount factors for promoting effective coordination in MARL
> and hence this has concentrated on discrete action spaces. However,
> based on the reviewer's advice we have tested the method on continuous
> action space, Water World Environment [2]. Additional
> details about this experiment are added in Section 4.5.1, page $18$ of
> the revised paper.
>
> We compare the results for continuous action space with the FACMAC
> approach presented in [3]. From the results, we found that
> the proposed DSDF approach provides for higher average returns when
> compared with the FACMAC approach This demonstrated the applicability of
> the proposed DSDF approach to environments with continuous action
> spaces.
>
> **Additionally it's unclear if the recommendation from \[1\] to train
> with multiple seeds was followed. From the description in text it seems
> only multiple runs were done for evaluation which is not enough
> especially given the added randomness.**
>
> Yes, we followed the multiple-seed approach while designing the
> experiments. The results for the SMAC environment are shown across $4$
> seeds. For the football environment, the results are shown for $6$
> seeds. Finally, for the lbForaging environment, the results are shown
> for $3$ seeds. For the case of the Water world environment, the results
> are shown for $2$ seeds. This additional information is added in
> captions of respective figures of the revised paper.
>
> **Some discussion discussing the distinction from ad-hoc teaming
> approaches in MARL would have been relevant and possibly better
> comparison baselines.**
>
> CTDE MARL approaches like the proposed DSDF or the base algorithms like
> QMiX typically rely on CTDE methodology whereby all the agents are
> trained together so that they develop the intuition of \"planning to
> coordinate\". During execution, a centralized \"master\" is not
> required. Ad-Hoc Teaming approaches (AHT) on the other hand is a
> framework that employs pre-trained self-interested agents to solve a
> coordinated task, where a \"Learner\" agent learns to set goals or
> incentives to drive the coordination. Additionally, AHT excels in
> circumstances where unseen agents may arrive in the system or may even
> change their existing type.
>
> In our work, we assume a decentralized execution framework whereby there
> is no central entity to understand that certain agents have degraded and
> are behaving erratically. This can only be understood by the discount
> factor hypernetwork by looking at the global states in conjunction with
> individual observations. Consequently, the appropriate discount factors
> \"dictate\" the behavior of individual agents to be short/far-sighted
> and thereby maximize total global reward.
>
> In the AHT scenario, we will need a centralized \"Learner\" with
> preferably global visibility which the given environments may not
> realistically offer. Secondly AHT learners can only incentivize the
> agents to do a task but it cannot \"dictate\" team-mates' behavior. Also
> typesetting all the variations of degraded actions may be a challenge.
> Finally, we are not sure if AHT Learners can influence the agents to be
> far or near-sighted as the agents have their own pre-existing policies.
>
> We would be pleased to include our point of view, as mentioned above, in
> the final document if given a chance. However given AHT is a slightly
> different track in agent coordination as compared to MARL, the AHT
> adoption for a degraded agent scenario may be left as future work
> following this one. We have added the discussion on this comparison in
> section $1$, page $2$ of the revised paper.

---

> > ### Author Response · Authors · 2023-03-15
> > **Respones to comments given by Reviewer DwDm (2/2)**
> >
> > **Lastly, during online adaptation with continuous degradation, it is
> > unclear if the paper still assumes access to global state information,
> > and if so, whether it still aligns with the CTDE paradigm.**
> >
> > During online adaptation, the discount factor network does not require
> > to be retrained whereas the agents' Q-networks and mixing network need
> > to be retrained (short adaptation phase) to fine-tune their
> > collaborative policy in accordance with the changed degraded scenario.
> > During such a re-training exercise, we utilize the global state
> > information and pass it to the agents. This is in accordance with the
> > CTDE principle where we utilize the global state information during
> > training. In essence, we use only the global state information when we
> > are updating the agents' policies.
> >
> >
> >
> > **Considering the primary contribution is another network to train a
> > scalar parameter, the method may introduce considerable complexity
> > without clear benefits.**
> >
> > The main contribution of the paper is on how to adjust the collaborative
> > policies of agents when some of the agents become noisy. The method
> > adjusts the discount factors ( and thereby far/nearsightedness) by which
> > the entire global coordination could be improved significantly. May we
> > point out that the approach of adjusting discount factors and its
> > effects in multi-agent coordination is rarely studied and given the
> > demonstrated results in both discrete and continuous action spaces, this
> > may be a promising area of further research. Another important facet is
> > that our method is computationally efficient and thus can be practically
> > implemented in Industry 4.0 scenarios in an energy-efficient manner.
> >
> > **Requested Changes**:
> >
> > **Use angle brackets $\langle$ for defining the Dec-POMDP tuple.**
> >
> > **Page 4: GLIE property Singh et al. (2000) should use $citep$.**
> >
> > **Page 5: Calling a factor is slightly confusing as it seems to be
> > subtracted rather than multiplied.**
> >
> > **More clarity on experiments**
> >
> > We made all the requested changes to the revised paper.
> >
> >
> > [1] Gottesman, Omer, Fredrik Johansson, Joshua Meier, Jack Dent, Donghun Lee, Srivatsan Srinivasan, Linying Zhang et al. "Evaluating reinforcement learning algorithms in observational health settings." arXiv preprint arXiv:1805.12298 (2018).
> >
> > [2] Gupta, Jayesh K., Maxim Egorov, and Mykel Kochenderfer. "Cooperative multi-agent control using deep reinforcement learning." In Autonomous Agents and Multiagent Systems: AAMAS 2017 Workshops, Best Papers, São Paulo, Brazil, May 8-12, 2017, Revised Selected Papers 16, pp. 66-83. Springer International Publishing, 2017.
> >
> > [3] Peng, Bei, Tabish Rashid, Christian Schroeder de Witt, Pierre-Alexandre Kamienny, Philip Torr, Wendelin Böhmer, and Shimon Whiteson. "Facmac: Factored multi-agent centralised policy gradients." Advances in Neural Information Processing Systems 34 (2021): 12208-12221.

---

### Review · Reviewer_cckB · 2023-03-04

**Summary Of Contributions:**

This work presents a new approach, called Deep Stochastic Discount Factor (DSDF), for multi-agent reinforcement learning (MARL) in the Centralized Training with Decentralized Execution (CTDE) paradigm. The method addresses the issue of random noise in agents, which can cause coordination uncertainty and lead to suboptimal team behavior. The authors propose tuning the discount factor for each agent uniquely based on the degree of degradation to mitigate this issue. The paper includes empirical results in SMAC, lbForaging and Google-Research Football domains, where DSDF is compared against existing methods.



**Audience:**

Yes

**Broader Impact Concerns:**

No impact concerns.

**Claims And Evidence:**

Yes

**Requested Changes:**

Please address the weaknesses described above.

**Strengths And Weaknesses:**

### Strengths
- The paper addresses an important problem in multi-agent reinforcement learning by considering scenarios in which an agent behaves differently from its policy due to degradation or aging.
- The authors provide empirical results on three different benchmarks to validate the proposed framework for dealing with noisy agents.

### Weaknesses

- The Iterative Penalization method assumes that we can observe if an agent performs a different action than instructed by its policy, which may not be realistic in real-world scenarios. I think this should just be a baseline with privileged knowledge used in experiments but not proposed as a new method.
- The choice of the penalization factor P in Method 1 is challenging, and the authors do not provide guidance on how to optimize it, which raises concerns about the applicability of this method.
- The paper could benefit from experiments that evaluate the generalization capabilities of agents beyond what they saw during training, considering that CTDE aims to train policies in a simulator (which should be perfect in theory) and test them in the real world (which is faulty and noisy).
- The authors only consider value-based MARL baselines, such as QMIX and IQL, and do not include policy-based baselines, such as MAPPO, which have demonstrated outstanding performance in CTDE domains. It would be informative to include these as baselines, or provide an explanation on whether DSDF can be extended to such methods, and if not, why?
- The absence of ablation studies makes it challenging to isolate the contribution of the proposed method.
- The writing can be improved, as some concepts are redundantly described, making the text difficult to read.

---

> ### Author Response · Authors · 2023-03-15
> **Respones to comments given by Reviewer cckB (1/2)**
>
> **The Iterative Penalization method assumes that we can observe if an
> agent performs a different action than instructed by its policy, which
> may not be realistic in real-world scenarios. I think this should just
> be a baseline with privileged knowledge used in experiments but not
> proposed as a new method.**
>
> We agree with the reviewer that monitoring the applied action is
> unrealistic in a real-time scenario. However, as we mentioned in the
> paper, the main contribution of the work is to arrive at the DSDF
> method. We mention the iterative penalization method as only a baseline
> to compare the results of the DSDF method. As said, this is just a
> baseline and not intended for real-time application. The text at the
> start of section $3.1$ of the revised paper has been appropriately
> modified to reflect this.
>
> **The choice of the penalization factor P in Method 1 is challenging,
> and the authors do not provide guidance on how to optimize it, which
> raises concerns about the applicability of this method.**
>
> The choice of the $P$ depends on the underlying environment for which
> collaboration is being learned. For example, the value of $P$ should be
> low for the cases where the agents take a longer time to converge and
> vice-versa. In the case of Football and SMAC environment, the value of
> $P$ is chosen to be $0.0001$, whereas for lbForaging the value of $P$ is
> chosen to be $0.01$. So, the choice involves a bit of domain knowledge.
> As discussed above this iterative penalization method just serves as a
> baseline. This information is added to the appendix of the revised
> paper.
>
> **The paper could benefit from experiments that evaluate the
> generalization capabilities of agents beyond what they saw during
> training, considering that CTDE aims to train policies in a simulator
> (which should be perfect in theory) and test them in the real world
> (which is faulty and noisy).**
>
> Yes, we agree the proposed method can be viewed as a form of
> generalization in reinforcement learning. However, this method focuses
> on agent generalization, at a limited scale, where agents can display a
> wide-ranging behavior discordant with policy. We assume the reviewer's
> suggestion is about testing the method for generalization abilities
> pertaining to agents adjusting to environment changes. We plan to
> address environmental generalization through similar techniques as part
> of future work. A short discussion on the generalization is added on
> Page $3$, Section $1$ in the revised version of the paper.
>
> **The authors only consider value-based MARL baselines, such as QMIX and
> IQL, and do not include policy-based baselines, such as MAPPO, which
> have demonstrated outstanding performance in CTDE domains. It would be
> informative to include these as baselines, or provide an explanation on
> whether DSDF can be extended to such methods, and if not, why?**
>
> In the revised paper, we have added results for another environment,
> Waterworld [1], in which action space is continuous. The
> choice of the environment is based on the suggestion of other reviewers.
> For this environment, we use a policy-based approach FACMAC
> [2] which is based on DDPG to train the individual agents
> (similar to MAPPO). Additional details about this experiment are added
> in Section 4.5.1, page 18 of the revised paper.
>
> From the results, we found that the proposed DSDF approach outperforms
> existing policy-based methods for Multi-Agent environments, like FACMAC.
> The choice of FACMAC is inspired by its closeness to the QMiX approach
> where both utilize a monotonic mixing network for arriving at $Q_{tot}$.
> Given the significant improvement in results over FACMAC, we are
> confident that it will also outperform policy-based algorithms like
> MAPPO.

---

> > ### Author Response · Authors · 2023-03-15
> > **Respones to comments given by Reviewer cckB (2/2)**
> >
> > **The absence of ablation studies makes it challenging to isolate the
> > contribution of the proposed method.**
> >
> > We compare the results of the proposed DSDF approach with vanilla QMiX
> > which does not have a provision to choose a dynamic discount factor
> > value. Also, we added a further comparison with the IQL method which has
> > a fixed discount factor value. From the results, it is evident that the
> > added feature of the dynamic discount factor makes up for all the
> > differences between the methods and the corresponding gain in
> > performance.
> >
> > **The writing can be improved, as some concepts are redundantly
> > described, making the text difficult to read.**
> >
> > We reviewed the document, modified the notations, and revised the text
> > to make it easily readable.
> >
> > [1] Gupta, Jayesh K., Maxim Egorov, and Mykel Kochenderfer. "Cooperative multi-agent control using deep reinforcement learning." In Autonomous Agents and Multiagent Systems: AAMAS 2017 Workshops, Best Papers, São Paulo, Brazil, May 8-12, 2017, Revised Selected Papers 16, pp. 66-83. Springer International Publishing, 2017.
> >
> > [2] Peng, Bei, Tabish Rashid, Christian Schroeder de Witt, Pierre-Alexandre Kamienny, Philip Torr, Wendelin Böhmer, and Shimon Whiteson. "Facmac: Factored multi-agent centralised policy gradients." Advances in Neural Information Processing Systems 34 (2021): 12208-12221.

---

### Author Response · Authors · 2023-03-15
**General answer to all reviewers**

We would like to thank all the reviewers for their reviews and
constructive feedback. We believe that the reviews have improved the
readability of the paper and the effectiveness of the proposed approach.
We have uploaded a revised version of the paper. All changes in the text
are colored in blue for easy identification. The main changes done in
the paper are:

1.  Added an experiment on the continuous action space environment
    (Water World) and compared the results with a policy-gradient
    approach known as FACMAC.

2.  Added some text on comparison of the proposed approach with ad-hoc
    teaming approaches (AHT) and generalization areas.

3.  All the notations are rechecked and revised accordingly for better
    readability.

4.  Redundant text from Section 3.1.1 is removed to make it crisper.

5.  The term "penalization factor" is changed to "penalization value" in the iterative penalization method section.

In addition, we performed an additional experiment to compare the
performance of using the same discount factor network (learned in
earlier experiments) versus that of the discount factor network trained
from scratch. The results are outlined in comments to the reviewer **Pdpm**.

We have included a comment-by-comment response to all reviewer's questions. Please find our comments below.

We are happy to continue the discussion and to further address any
questions or requests the reviewers might have.

---

### Decision · Action_Editors · 2023-04-04

**Recommendation:** Reject

**Comment:**

This paper addresses the problem of agents’ degradation in centralised training with decentralised execution (CTDR) setting of multi-agent reinforcement learning. It is assumed that this degradation translates in unwanted and uncontrolled stochasticity of the individual policies. The proposed approach, DSDF, consists in learning the discount factor of each individual agent, the intuition being that associating a lower discount factor to an unreliable agent may be globally beneficial.

All reviewers found the topic to be of interest for the TMLR audience, but none support accepting the paper, even after rebuttal and revision. The remaining concerns notably encompass:
* the required assumptions (full observability during the adaptation stage, unclear required non-stochasticity of the system)
* missing experiments, especially regarding generalisation
* the soundness of the proposed approach (roughly, what is the learning signal for the discount factor, what mixing Q-values with various discount actually does)
* overall clarity of the paper (eg, $Q_\text{tot}$ is never defined, the target network of Q_mix seems to disappear without explanation, numerous typos such as missing square in Eq 4, unclear or ill-defined notations, etc, all them making the paper pretty hard to assess)

**Audience:**

Yes

**Claims And Evidence:**

No